# GIFT: A Framework Towards Global Interpretable Faithful Textual Explanations of Vision Classifiers

**Éloi Zablocki**[*,1]                                            *eloi.zablocki@valeo.com*
**Valentin Gerard**[*,1,†]                                        *valentin.gerard@epfl.ch*
**Amaia Cardiel**[1,2]                                            *amaia.cardiel@valeo.com*
**Eric Gaussier**[2]                             *eric.gausier@univ-grenoble-alpes.fr*
**Matthieu Cord**[1,3]                                           *matthieu.cord@valeo.com*
**Eduardo Valle**[1,‡]                                        *eduardo.valle@intercom.io*

[*] *equal contribution*
[1] *Valeo.ai, Paris, France*
[2] *Université Grenoble Alpes, France*
[3] *Sorbonne Université, Paris, France*
[†] *now at EPFL ;* [‡] *now at Intercom*

**Reviewed on OpenReview:** *https://openreview.net/forum?id=OwhW5MpFmD*

## Abstract

Understanding the decision processes of deep vision models is essential for their safe and trustworthy deployment in real-world settings. Existing explainability approaches, such as saliency maps or concept-based analyses, often suffer from limited faithfulness, local scope, or ambiguous semantics. We introduce **GIFT**, a post-hoc framework that aims to derive *Global*, *Interpretable*, *Faithful*, and *Textual* explanations for vision classifiers. GIFT begins by generating a large set of faithful, local visual counterfactuals, then employs vision–language models to translate these counterfactuals into natural-language descriptions of visual changes. These local explanations are aggregated by a large language model into concise, human-readable hypotheses about the model's global decision rules. Crucially, GIFT includes a verification stage that quantitatively assesses the *causal effect* of each proposed explanation by performing image-based interventions, ensuring that the final textual explanations remain faithful to the model's true reasoning process. Across diverse datasets, including the synthetic CLEVR benchmark, the real-world CelebA faces, and the complex BDD driving scenes, GIFT reveals not only meaningful classification rules but also unexpected biases and latent concepts driving model behavior. Altogether, GIFT bridges the gap between local counterfactual reasoning and global interpretability, offering a principled approach to causally grounded textual explanations for vision models.

## 1 Introduction

Explainability is crucial for deploying deep vision models in high-stakes applications such as autonomous driving (Omeiza et al., 2022; Zablocki et al., 2022) and medical imaging (Tjoa & Guan, 2019), where understanding model decisions ensures trust and safety. Solutions for explainability include feature attribution methods (Selvaraju et al., 2017; Sundararajan et al., 2017), concept-based approaches (Kim et al., 2018; Fel et al., 2023b), and model-surrogate methods (Ribeiro et al., 2016; Lundberg & Lee, 2017). However, those approaches often lack faithfulness, with explanations confounded by spurious correlations in the data without causal link to the model's decision. As a result, they fail to accurately reflect the model's true reasoning process, leading to potentially misleading interpretations (Rudin, 2019; Dasgupta et al., 2022). Counterfactual explanations (Wachter et al., 2017) address that limitation by identifying minimal input changes that alter model outputs, capturing causal relationships. However, counterfactual explanations have

Table 1: **_Post-hoc_ methods for explaining vision classifiers.**

| Explainability family | Scope | Level of output interpretability | Faithful to the model |
|---|---|---|---|
| Input attribution | Local | Low (saliency maps) | No |
| Model-surrogate | Local | Average (surrogate model) | No |
| Concept-based | Global | Low (CAV) or High (text) | No |
| Counterfactual | Local | Average (images) | Yes |
| **GIFT** (ours) | Global | High (text) | Yes |

their own limitations: they are inherently local, focusing on specific instances; often are hard to interpret, as they depend on visual analysis of the generated changes; and can be ambiguous, with a single counterfactual modification potentially stemming from multiple plausible causes.

In this work, we address those limitations by introducing GIFT: a framework for obtaining more Global, Interpretable, Faithful, and Textual explanations for deep vision models. GIFT builds upon counterfactual explanations while mitigating their weaknesses through three key innovations: (1) *Global Reasoning.* To move beyond the locality of individual counterfactuals, GIFT aggregates multiple counterfactual explanations across the model's input domain. This enables discovering broader, global insights into the model's behavior. (2) *Natural Language Interpretability.* GIFT leverages natural language to transform raw counterfactual explanations into clear, human-readable descriptions. By utilizing the reasoning capabilities of Large Language Models (LLMs) (Radford et al., 2019), GIFT resolves ambiguities in counterfactuals and organizes them into coherent global explanations. (3) *Verification of Explanations.* GIFT includes a novel verification step that measures the *causal effect* of candidate explanations. By intervening on images using image-editing models, GIFT ensures that the derived explanations are faithful to the model being explained.

As a result, the explanations produced by GIFT are global, textual, disambiguated, and supported by causal verification to the underlying model. Unlike a standalone model, GIFT is a framework that can be instantiated with recent models for counterfactual generation, (vision-)language reasoning, and text-guided image editing. That flexibility is essential for handling the diverse data domains and decision models across different tasks, as general-purpose models often struggle with the specificity required in certain domains.

We validate GIFT across diverse binary-classification scenarios: on the CLEVR dataset (Johnson et al., 2017), GIFT uncovers classification rules in a controlled setting with complex compositionality; on the CelebA dataset (Liu et al., 2015), GIFT discovers fine-grained relationships between data features and classification outcomes; and on the BDD-OIA dataset (Yu et al., 2020), GIFT identifies biases in a challenging domain with driving scenes. Our contributions are:

- We introduce the first framework for global, textual, and counterfactual explanations for vision classifiers. The faithfulness of the explanations is supported by causality quantification.

- We derive global explanations by combining two ideas: (1) gathering counterfactual signals across the model's input domain, which are inherently causal although very local; and (2) reasoning with an LLM to uncover global insights from those local signals. Both ideas and their combined synergy are novel, to our knowledge.

- We analyze two complementary causal metrics, demonstrating their relationship, and providing GIFT with verification tools to measure causal association.

- We validate GIFT's ability to generate meaningful global explanations across several natural image domains and use cases in binary classification.

## 2 Related Work

Explanation methods are typically categorized as *intrinsic* (built into models during training for inherent interpretability (Hendricks et al., 2016; Zhang et al., 2018)) or *post-hoc* (applied post-training to explain

decisions (Zhang et al., 2018; Chen et al., 2019)). As shown in Tab. 1, post-hoc methods further divide into *local* (per-instance) and *global* (model-level).

**Feature Attribution Methods** highlight regions of the input image that most influence the model's predictions, often visualized as a saliency map. Gradient-based approaches (Zeiler & Fergus, 2014; Selvaraju et al., 2017; Sundararajan et al., 2017; Fel et al., 2021; Wang et al., 2024b) back-propagate gradients to identify influential features, and perturbation-based methods remove or alter parts of the input to observe resulting changes in model output (Fong & Vedaldi, 2017; Wagner et al., 2019; Bacha & George, 2025). Though widely used, attribution maps offer only local, instance-level explanations. They require user interpretation, introducing subjective biases (Borowski et al., 2021), and can be unreliable, sometimes resembling edge-detectors on CNNs.

**Model-Surrogate Methods** explain black-box models through interpretable surrogate models (Engel et al., 2024; Páez, 2024). Approaches such as LIME (Ribeiro et al., 2016) and SHAP (Lundberg & Lee, 2017) are widely used to provide *local* explanations, with some attempts at *global* surrogates for more holistic interpretability (Frosst & Hinton, 2017; Zilke et al., 2016; Harradon et al., 2018). These methods simplify inputs, e.g., using superpixels (Ribeiro et al., 2016) or high-level input features (Lundberg & Lee, 2017), which can result in reduced fidelity and limit the reliability of explanations for complex, high-dimensional data (Zhang et al., 2023).

**Counterfactual Explanations**, in the context of image classification, are images minimally altered to yield a different classification outcome, thus revealing semantic changes needed to cross the classifier's decision boundary (Wachter et al., 2017). Retrieval-based (Hendricks et al., 2018; Goyal et al., 2019b; Vandenhende et al., 2022) and generator-based methods (with GAN (Rodríguez et al., 2021; Jacob et al., 2022; Zemni et al., 2023) or diffusion models (Jeanneret et al., 2022; Augustin et al., 2022; Jeanneret et al., 2023; Sobieski & Biecek, 2024)) are used to create realistic counterfactuals. While counterfactuals can provide intuitive insights, they are inherently local explanations. Moreover, they demand careful user interpretation, which may introduce human biases (Borowski et al., 2021; Zemni et al., 2023).

**Concept-based Methods** explain model decisions through human-interpretable 'concepts' (Kim et al., 2018; Lee et al., 2023), from neuron-level analyses (Bau et al., 2017) to layer-level approaches such as CAVs (Kim et al., 2018). Most require annotated data for concept presence/absence (Kim et al., 2018), though some extract concepts unsupervised via matrix factorization (Zhang et al., 2021; Kumar et al., 2021; Fel et al., 2023b). These methods typically require model weights, limiting applicability, and are architecture-specific, with most methods only supporting CNN-based models (Kumar et al., 2021; Kamakshi et al., 2021; Fel et al., 2023b). Manually defined concepts may overlook biases, while unsupervised ones still require human interpretation (Fel et al., 2023a). In contrast, GIFT uses counterfactual signals instead of predefined concepts, avoiding prior biased assumptions. Its textual explanations are directly interpretable and verifiable for faithfulness, providing more reliable insights than correlation-based approaches. Similarly, MAIA (Shaham et al., 2024) employs a multimodal agent to automate interpretability experiments by composing tools for neuron description and bias discovery, though it relies heavily on iterative hypothesis testing rather than aggregating counterfactual evidence.

**Error Explanations and Shortcut Mitigation Methods** uncover model failure modes using natural language. Some rely on user- or LLM-generated inputs (Abid et al., 2022; Csurka et al., 2024; Prabhu et al., 2023; Wang et al., 2024a), which can bias the scope of discovered errors. Others identify systematic failures through data partitioning and labeling (Eyuboglu et al., 2022; Wiles et al., 2022), but require ground-truth labels. In contrast, GIFT uses the LLM solely to summarize counterfactual evidence—not as a source of domain knowledge—thereby reducing LLM-induced bias. It also goes beyond failure analysis to explain both correct and incorrect predictions. Kuhn et al. (2025) propose an unsupervised shortcut detection and mitigation pipeline for transformers that clusters patch key-space, extracts prototypical patches, generates textual prototype descriptions with a VLM, and mitigates shortcuts via patch ablation and last-layer retraining. While complementary to our aims, their method operates on model internals and focuses on ablation-based mitigation tailored to ViT architectures; by contrast, GIFT is counterfactual-centered and model-agnostic, grounding explanations in generated edits and verifying them with causal measurements.

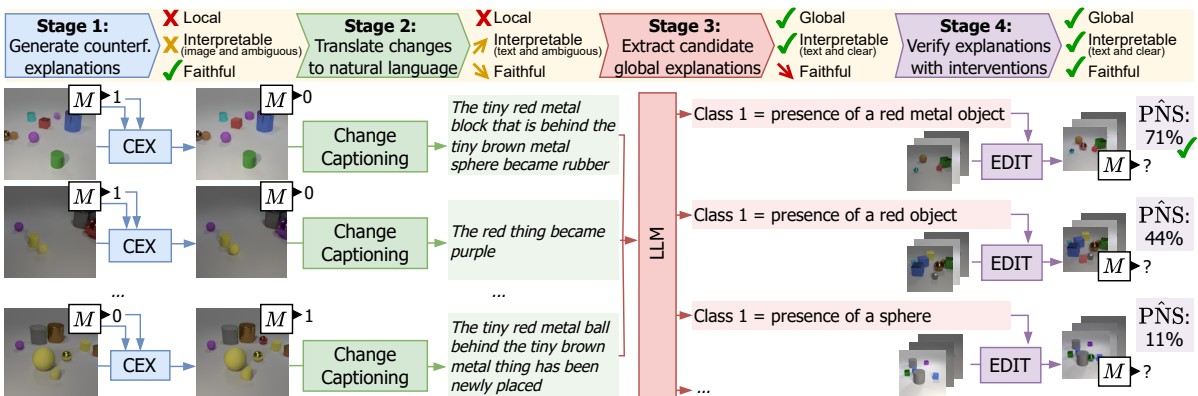

Figure 1: **Overview of GIFT**. Given a classifier $M$ (here discriminating images with a 'red metal object'), GIFT extracts explanations in four stages: Stage 1 generates local visual counterfactual explanations for several images. The counterfactuals are by nature faithful to the classifier as they reveal semantic and minimal changes to the query images that flip the classifier's output. Stage 2 translates in natural language the visual differences between original and counterfactual images with an image change captioning model ; this enhances interpretability but risks introducing potential noise. Stage 3 applies an LLM to aggregate local explanations into candidate global explanations ; this disambiguates local evidences. Lastly, Stage 4 filters out or validates these global explanations with intervention studies, to ensure faithfulness with respect to the classifier.

## 3 GIFT Framework

GIFT automatically identifies and validates explanations for a differentiable classifier $M$. An *explanation*, here, describes features, attributes, or concepts that are significant to the classifier's decision. Explanations should be meaningful to humans and faithful to the classifier's behavior.

**Overview.** GIFT follows four stages (Fig. 1). In Stage 1 (Sec. 3.1), it creates local explanations by generating several counterfactual input-pairs, highlighting faithful visual features relevant to $M$'s decision at the individual input level. In Stage 2 (Sec. 3.2), it uses a Vision Language Model (VLM) to translate the differences between each image in a counterfactual pair into interpretable text descriptions. In Stage 3 (Sec. 3.3), GIFT employs an LLM to identify recurrent patterns on those local explanations to obtain global candidate explanations. Finally, in Stage 4 (Sec. 3.4), it evaluates the candidate explanations on causal metrics, measuring their faithfulness to $M$'s decision process.

### 3.1 Stage 1: Faithful visual and local explanations

GIFT begins by producing *local, faithful visual explanations* for the target model $M$ through counterfactual image generation. Counterfactual explanation methods (Zemni et al., 2023; Jeanneret et al., 2023) search for minimal, semantically meaningful modifications of an input image $x$ that flip the model's prediction. Unlike adversarial attacks (Szegedy et al., 2014), which often rely on imperceptible perturbations, counterfactuals aim to expose the model's *semantic decision boundaries* by altering visual features that the model truly relies on (Freiesleben, 2022). By explicitly requiring the model's output to change, these methods are faithful by definition: they do not rely on surrogate approximations but instead directly probe the model's decision boundary. This ensures that the resulting explanations capture a causal link to $M$'s decision.

Concretely, given a sample set $\mathcal{I}$ of $N$ input images, we use a counterfactual generator CEX that directly accesses the model $M$ and its decision $M(x)$ to synthesize a counterfactual $x'$ such that $M(x') \neq M(x)$. This yields a set of image pairs:

$$\mathcal{P} = \{(x, x') \mid x \in \mathcal{I}, \, x' = \text{CEX}(x, M(x), M)\}. \tag{1}$$

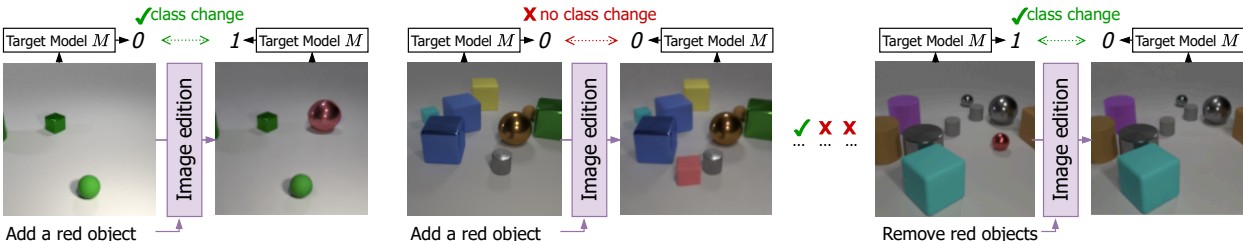

Figure 2: **Causal interventions, Stage 4.** For a candidate explanation $e$ (e.g., 'class 1 = presence of a red object'), we use an image-editing model to add or remove the underlying concept $c_e$ (e.g., 'red object') and observe the impact on the classification outcome, which is aggregated to compute the CaCE (Eq. 3) and $\widehat{\text{PNS}}$ (Eq. 4). In the example, the classifier $M$ recognizes images with a 'red metal object', and we observe, as expected, a partial causal effect: removing red objects impacts the outcome, but inserting non-metal red ones does not.

Each pair $(x, x')$ provides a faithful, instance-specific view of the local visual change responsible for a decision flip. These local counterfactuals constitute the atomic evidence from which subsequent stages of GIFT will infer higher-level, global explanations.

### 3.2 Stage 2: From visual counterfactuals to text

For each counterfactual pair, GIFT creates a change caption, translating the visual changes from the original image to the counterfactual into simple, descriptive natural language.

Visual counterfactuals, *per se*, require tedious and subjective manual comparison to become interpretable. Moreover, natural language is often more accessible to humans than low-level visual cues, such as saliency maps or feature vectors (Sammani et al., 2022; Nauta et al., 2023). We use instead vision-language models (VLMs) to *change-caption* (Guo et al., 2022; Qiu et al., 2020; Hosseinzadeh & Wang, 2021) each image pair, i.e., to automatically describe the changes between each original image $x$ and its counterfactual $x'$.

Concretely, we use a Change Captioning model CC that takes as input the pair of images and outputs a change caption $t = \text{CC}(x, x')$. While more interpretable than visual comparisons, these captions remain local to each image pair and may be less faithful due to the compressed nature of text and potential noise introduced by the VLM. The next stages address these.

### 3.3 Stage 3: Candidate global explanations

GIFT will now gather all change captions, analyze them for recurrent patterns, and propose candidate global explanations for the target model behavior. Concretely, we gather the set of all local explanation tuples $\mathcal{T} = \{(\text{CC}(x, x'), M(x), M(x')) | (x, x') \in \mathcal{P}\}$ and use an LLM to summarize them into a set of candidate global explanations $\mathcal{E} = \text{LLM}(\mathcal{T})$ that helps explaining the classifier's behavior. Remark that the LLM has access, for each tuple, to both the change caption and to what happened to the model decision $(M(x) \rightarrow M(x'))$. The LLM has no direct access to the model itself.

This stage addresses the shortcomings of the local explanations. It *disambiguates local evidence* where a given counterfactual change implies multiple plausible explanations. For example, a 'red metal ball became brown' explanation, in isolation, could mean that the classifier tracks the absence of 'red objects' (or 'red metal objects') in class 0 or the presence of 'brown objects' (or 'brown metal objects') in class 1. This stage also *filters out noise*, alleviating linguistic variations and eliminating most irrelevant or inconsistent explanations raised in Stage 2.

### 3.4 Stage 4: Hypotheses verification

The initial explanations at Stage 1, although local and laborious to interpret, are faithful. By Stage 3, we have global interpretable explanations but cannot guarantee their faithfulness, as the VLM and LLM

may introduce noise. This challenge is precisely what motivates Stage 4, that verifies which explanations in $\mathcal{E}$ remain faithful to the classifier. *Faithfulness* in GIFT is quantified causally rather than correlationally. Specifically, a global explanation is retained only if interventions on the identified concept significantly alter the classifier's decision. This ensures that accepted explanations correspond to necessary and sufficient *causal* factors. For each explanation $e \in \mathcal{E}$ (e.g., 'class 1 implies the presence of a red object'), we extract the underlying concept $c_e$ (e.g., 'red object') and apply a two-step verification using these concepts.

**Coarse filter.** This first step measures the *correlation* between the model's decision and the underlying concept $c_e$ of each candidate explanation $e \in \mathcal{E}$. We use the Directed Information (DI) (Gallager, 1968) between the concept $c_e$ and the predicted class label $y$ as the correlation metric $\mathrm{DI}(c_e, y) = \frac{I(c_e; y)}{H(c_e)}$ where $I(c_e; y)$ is the mutual information between the concept presence and the classifier output, and $H(c_e)$ is the entropy of the concept presence. DI quantifies how much the presence of concept $c_e$ explains the classifier's decision $y$, assessing the directional influence of the underlying explanation on the classification outcome. To estimate $c_e$'s presence in an image $x$, we use a Visual Question Answering model $\mathrm{VQA} : (x, c_e) \to \{0, 1\}$.

We measure the DI for each explanation, rank them, and keep the most promising ones. This pre-filter is motivated by the fact that applying the VQA model provides a correlation metric that is much easier and computationally cheaper to obtain than causal metrics requiring image interventions.

**Fine filter.** We evaluate high-correlation explanations $e \in \mathcal{E}$ that pass the coarse filter on their *causal effect* on the classifier's decision by measuring the causal association (Pearl, 2009; Goyal et al., 2019a) between their underlying concept $c_e$ and the model decision $y$. We partition the samples of a validation set of images according to $c_e$, *intervene* on those images by inserting or removing $c_e$ and verify the impact of those interventions on the model decision, as illustrated in Fig. 2.

Concretely, we use $\mathrm{VQA}(x, c_e)$ to partition a validation set $\mathcal{X}$ into images where $c_e$ is present ($\mathcal{X}_{c_e=1}$) or absent ($\mathcal{X}_{c_e=0}$). We then use a text-guided image-editing model EDIT to intervene and edit $x$ into $\tilde{x} = \mathrm{EDIT}(x, c_e)$, flipping the presence of $c_e$ in the image. Specifically, we create images $x^{+c_e} := \mathrm{EDIT}(x, c_e)$ for each $x \in \mathcal{X}_{c_e=0}$ by adding $c_e$, (similarly, $\forall x \in \mathcal{X}_{c_e=1}, x^{-c_e} := \mathrm{EDIT}(x, c_e)$). Image editing falls short of the unfeasible ideal of modifying physical reality in past image acquisition and assumes other concepts will be left relatively undisturbed. Nevertheless, this practical procedure has shown good results in estimating the actual causal effect (Goyal et al., 2019a). We also assume that edited images remain in-distribution for the classifier $M$. We now present two metrics to evaluate the causal effect of the interventions.

*Causal Concept Effect (CaCE)*: Goyal et al. (2019a) defines $\mathrm{CaCE}_{c_e}$, for a given concept $c_e$, as:

$$\mathrm{CaCE}_{c_e} = \mathbb{E}[M(x)|c_e = 1] - \mathbb{E}[M(x)|c_e = 0]. \tag{2}$$

$\mathrm{CaCE}_{c_e}$ measures how the addition or removal of $c_e$ induces any class change. Specifically, for binary classifiers or multi-class classifiers evaluated in a one-versus-all manner, considering $M(x) \in \{0, 1\}$, we have $\mathrm{CaCE}_{c_e} \in [-1, 1]$, which can be estimated by:

$$\mathrm{CaCE}_{c_e} = \frac{1}{|\mathcal{X}|}\left( \sum_{x \in \mathcal{X}_{c_e=1}} M(x) - M(x^{-c_e}) + \sum_{x \in \mathcal{X}_{c_e=0}} M(x^{+c_e}) - M(x) \right), \tag{3}$$

with the following interpretation:

- If $\mathrm{CaCE}_{c_e}$ is sufficiently positive (resp. sufficiently negative), then the presence of $c_e$ 'entails' class 1 (resp. 0) and its absence 'entails' class 0 (resp. 1),

- Otherwise, $c_e$, in isolation, has a negligible impact on the classification decision.

*Probability of Necessary and Sufficient Cause (PNS)*: Based on Tian & Pearl (2000), we estimate the probability $\mathrm{PNS}_{c_e, y}$ of $c_e$ to be both a necessary and a sufficient cause for the class $y$. As shown in Sec. C.2, it

can be estimated by:

$$\widehat{\text{PNS}}_{c_e,y} = \frac{1}{|\mathcal{X}|} \left( \sum_{x \in \mathcal{X}_{c_e=1}} \mathbb{1}[M(x^{-c_e}) \neq y \wedge M(x) = y] + \sum_{x \in \mathcal{X}_{c_e=0}} \mathbb{1}[M(x^{+c_e}) = y \wedge M(x) \neq y] \right). \quad (4)$$

In addition, for {0,1}-class settings, we have: $\text{CaCE}_{c_e} = \widehat{\text{PNS}}_{c_e,y=1} - \widehat{\text{PNS}}_{c_e,y=0}$. Thus, $\text{CaCE}_{c_e}$ and $\widehat{\text{PNS}}_{c_e,y=1}$ (or equivalently $\text{CaCE}_{c_e}$ and $\widehat{\text{PNS}}_{c_e,y=0}$) bring complementary information as $\widehat{\text{PNS}}_{c_e,y=1}$ better evaluates the impact of $c_e$ on $y = 1$, while $\text{CaCE}_{c_e}$, depending on its closeness to $\widehat{\text{PNS}}_{c_e,y=1}$, reveals how unidirectional the causal effect of $c_e$ is on $y = 1$ (details in Sec. C.3). In the remainder of the paper, we write $\widehat{\text{PNS}}$ for $\widehat{\text{PNS}}_{c_e,y=1}$ and CaCE for $\text{CaCE}_{c_e}$.

### 3.5 Further exploration of the explanation space.

Ideally, a single explanation with a very high CaCE and $\widehat{\text{PNS}}$ would fully capture the model's decision rule. However, for complex classifiers, single explanations rarely suffice due to entangled feature interactions. GIFT allows end-users to explore and refine explanations by proposing new ones or combining them with those identified by the framework. These refined explanations are then re-evaluated with CaCE and $\widehat{\text{PNS}}$ to assess their causal impact on the model's output. As GIFT's intermediate stages are fully interpretable, users can stay in the loop, enabling an iterative process to deepen their understanding of the model's decision-making.

## 4 Experiments

We evaluate GIFT on three use cases of increasing complexity. In Sec. 4.1, we test GIFT's ability to uncover binary classification rules in a controlled setting with the CLEVR dataset. In Sec. 4.2, we show how GIFT finds fine-grained explanations for a binary classifier trained on CelebA. Finally, in Sec. 4.3, we use GIFT to reveal biases in a model trained on BDD-OIA, a dataset of complex driving scenes.

### 4.1 Uncovering classification rules on CLEVR

Here, the goal is to uncover the meaning of *obfuscated labels* ('0' or '1') with GIFT explanations.

**Data and target classifiers.** We use the CLEVR dataset (Johnson et al., 2017), which presents synthetic but photorealistic arrangements of objects with various shapes, colors, and textures on a neutral background. We train various target classifiers on binary tasks to recognize a specific visual rule such as 'cyan object present' or 'yellow rubber object present'. CLEVR images hide complex compositional potential behind their minimalist appearance, with diverse colors, textures, and shapes. Uncovering the underlying rule in each classifier from local examples is challenging, even for humans (see Sec. F.1). We include both ViT-Small-Patch16-224 (Dosovitskiy et al., 2021) and ResNet-34 (He et al., 2016) architectures. Our evaluation comprises 12 unique combinations of visual rule × architecture.

**Instantiation of GIFT.** The components for each stage are based on the dataset's specific characteristics. For the counterfactual generation (Stage 1), we use OCTET (Zemni et al., 2023) with BlobGAN (Epstein et al., 2022), as it is specifically designed for compositional scenes. CLIP4IDC (Guo et al., 2022), handles change-captioning (Stage 2) as it was trained on the same visual domain (Park et al., 2019). For Stage 3, we experiment with either ChatGPT4 (OpenAI, 2023) or Qwen2.5-72B-Instruct (Yang et al., 2024). Interventions (Stage 4) use a Stable Diffusion model (Rombach et al., 2022) adapted to CLEVR for targeted addition and removal of objects (Cho et al., 2024). The VQA model (Stage 4) is MiniCPM (Yao et al., 2024).

**Main results.** Tab. 2 reports the trials of GIFT uncovering the hidden visual rule underlying the target classifiers after Stages 1–4. The true visual rule was uncovered in 11/12 cases, with both ChatGPT or Qwen used in Stage 3. We consider the trial successful when the true rule appears on top after Stage 4 and ranking by the causal metrics.

We provide a detailed qualitative analysis of the 'Red metal object' (ViT) trial in Fig. 1. (Stage 1:) Counterfactual image generation flips the model's decision by modifying objects' presence, location, or appearance.

| Rule (Ground Truth) | Arch. | GIFT GPT | GIFT Qwen | Abl. | CaCE | $\widehat{PNS}$ |
|---|---|---|---|---|---|---|
| Cyan object | ResNet | ✓ | ✓ | ✗ | 62.2 | 62.2 |
| Purple object | ResNet | ✓ | ✗ | ✓ | 71.2 | 71.2 |
| Metal object | ResNet | ✓ | ✓ | ✓ | 24.2 | 24.2 |
| Rubber object | ResNet | ✓ | ✓ | ✗ | 29.8 | 29.8 |
| Red metal object | ResNet | ✗ | ✓ | ✗ | N/A | N/A |
| Yellow rubber object | ResNet | ✓ | ✓ | ✗ | 67.2 | 67.7 |
| Cyan object | ViT | ✓ | ✓ | ✗ | 63.1 | 63.1 |
| Purple object | ViT | ✓ | ✓ | ✓ | 70.2 | 70.2 |
| Metal object | ViT | ✓ | ✓ | ✓ | 37.4 | 37.4 |
| Rubber object | ViT | ✓ | ✓ | ✗ | 30.8 | 31.3 |
| Red metal object | ViT | ✓ | ✓ | ✗ | 70.7 | 70.7 |
| Yellow rubber object | ViT | ✓ | ✓ | ✗ | 71.7 | 71.7 |

Table 2: **Classification rule reverse engineering on CLEVR.** Success criterion is to have the hidden true rule on top after ranking them by the CaCE and PÑS causal metrics (Eq. 3, 4) after Stages 1–4. GIFT succeeds for all but one case with both metrics. An ablation (Abl.) with *independent* captions instead of *change* captions misses most rules (see App. F.4). Metrics in %.

| Concepts associated to the class 'Old' | DI | CaCE | $\widehat{PNS}$ |
|---|---|---|---|
| *after Stage 3* | | | |
| Receding Hairline | 29.9 | 2.0 | 5.0 |
| Neck Wrinkles | 28.1 | 3.0 | 3.5 |
| Wrinkles on Forehead | 26.4 | 6.5 | 7.0 |
| Wrinkles around Eyes | 20.7 | 6.5 | 7.0 |
| Glasses | 16.5 | 11.5 | 16.0 |
| Drooping Eyelids | 15.4 | 2.5 | 4.0 |
| ... | ... | ... | |
| Detailed Background | 0.2 | | |
| Low Camera Angle | 0.1 | | |
| *after manual combinations of concepts with* $\widehat{PNS} \geq 5\%$ *(Sec. 3.5)* | | | |
| Hairline + Wrinkles on Forehead | 47.8 | 8.2 | 8.2 |
| Glasses + Wrinkles around Eyes | 60.4 | 21.7 | 23.6 |
| Glasses + Hairline | 61.4 | 24.6 | 25.4 |
| Glasses + Hairline + Wrinkles on Forehead | 78.2 | 26.0 | 27.1 |
| Glasses + Wrinkles on Forehead | 59.6 | 24.1 | 27.6 |
| Glasses + Wrinkles on Forehead + Eyes | 65.6 | 26.8 | 28.9 |

Table 3: **Output of GIFT on the CelebA 'Old' classifier.** Concepts are evaluated with causal metrics when DI $\geq$ 15%. The full table is given in App. G. DI, CaCE and $\widehat{PNS}$ in %.

(Stage 2:) Because each counterfactual pair provides a very local and ambiguous explanation, each change caption extracted has weak evidence. There is also some linguistic variation. (Stage 3:) The summarization has to filter out the noise and recover robust trends in the whole sample of captions and proposes several potential explanations. Explanation verification (Stage 4) turns out to be critical to uncover the true rule. Without Stage 4 and ranking by causal metric (Sec. 3.4) the user has no guidance to distinguish them.

The use of two causal metrics is informative since close results indicate that the causal effect is unidirectional, meaning that the evaluated rule is either the true one or an overspecification of the true one (such as 'Red metal object' instead of 'Red object'). Indeed, in this use case, the true (Tab. 2) and the overspecified (Tab. 6 in Supp. Mat.) rules are always such that $|\widehat{PNS} - CaCE| \leq 0.5\%$ which suggests that appropriate thresholding can reduce the amount of candidate rules.

The 'Red metal object' (ResNet-34) failure case illustrates how users can interact with GIFT. As shown in Tab. 6 (in Supp. Mat.), the LLM at Stage 3 generates partial rules 'Red object' and 'Metal object', which Stage 4 tags with a CaCE of 42.9% (vs $\widehat{PNS}$ = 43.9%) and 11.6% (vs $\widehat{PNS}$ = 13.6%) respectively. When we manually try the combination of these two partial rules, we obtain the true rule 'Red Metal Object' (CaCE=$\widehat{PNS}$=61.6%). Full details are in Sec. F.4.

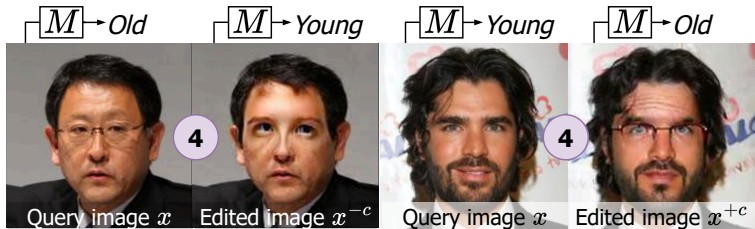

Figure 3: **Samples from the intervention study on the CelebA-'Old' classifier**. The combined concepts under scrutiny are: 'Glasses', 'Wrinkles on Forehead', 'Wrinkles around Eyes'. Each pair has the query image on the left and the edition on the right.

## 4.2 Exploring detailed explanations on CelebA

**Data and target classifier.** We evaluate GIFT on CelebA (Liu et al., 2015), a dataset of human faces that introduces real-world challenges. Following Singla et al. (2020); Jacob et al. (2022), the target classifier is a DenseNet121 (Huang et al., 2017) trained to predict whether a face is 'Old' or 'Young'.

**Instantiation of GIFT.** Unlike in the CLEVR use-case, where change-captioning and image-editing models were specifically tailored to the dataset, this use-case employs more general-purpose models. Counterfactual generation (Stage 1) uses the diffusion-based model ACE (Jeanneret et al., 2023). Change captioning (Stage 2) employs the Pixtral-12B VLM (Agrawal et al., 2024), guided by a one-shot in-prompt annotated sample as guidance. Concept identification (Stage 3) is performed zero-shot using Qwen 2.5 72B Instruct, quantized to 8 bits (Yang et al., 2024). Interventions (Stage 4) use a combination of an image-conditioned version of Stable Diffusion 3 (Esser et al., 2024) and an in-painting version of Stable Diffusion 2 (Rombach et al., 2022) relying on masks obtained with Florence-2 (Xiao et al., 2024). Details in App. G. Lastly, as causal metrics provide different rankings on this dataset, we favor $\widehat{\text{PNS}}$.

**Main results.** Tab. 3 shows a subset of explanatory concepts proposed by GIFT. It includes reasonable attributes, such as 'Neck Wrinkles' and 'Receding Hairline', and more unexpected ones, such as 'detailed background'. Directed information (DI) scores provide an initial filter to remove explanations whose concepts have no significant relationship to the classifier's output, allowing us to focus on more promising hypotheses. From that refined set, we conduct intervention studies to measure causal effects. Overall, concepts derived from Stage 3 have low causal metrics, despite high DI scores: intervening on any of these attributes alone does not change the classifier's decision. This suggests that the classifier is robust and not overly reliant on single attributes. Yet, 'Glasses' is an exception, with 16% probability of being a necessary and sufficient cause to the 'Old' class, likely reflecting a training data bias, as 'Glasses' correlates with 'Old' at over 20%.

As described in Sec. 3.5, we further explore the explanations by combining and reevaluating concepts with $\widehat{\text{PNS}} \geq 5\%$ (Tab. 3, rows '*after manual combination ...*'). Some concepts that, individually, have low causal metrics yield much higher values after combination, indicating an increased understanding of the classifier. For example, 'Hairline' and 'Glasses' have respective $\widehat{\text{PNS}}$ of 5 and 16% but reach 25.4% when combined. Fig. 3 illustrates the targeted intervention for the combined attributes 'Glasses', 'Wrinkles on Forehead', and 'Wrinkles around Eyes', which is sufficient to flip the model's decision in the two query images while leaving the rest of the face mostly unchanged.

## 4.3 Discovering classifier bias on BDD-OIA

Here, the task is to identify any classifier's rules, including spurious or unexpected ones.

**Data and target classifier.** We evaluate GIFT on BDD-OIA (Xu et al., 2020), a dataset of front-camera driving scenes in complex urban environments annotated with admissible car actions (turn left, go ahead, turn right, stop/decelerate). Widely used to benchmark explanation methods (Jacob et al., 2022; Zemni et al., 2023; Jeanneret et al., 2024), BDD-OIA is paired with a biased binary classifier for the 'turn

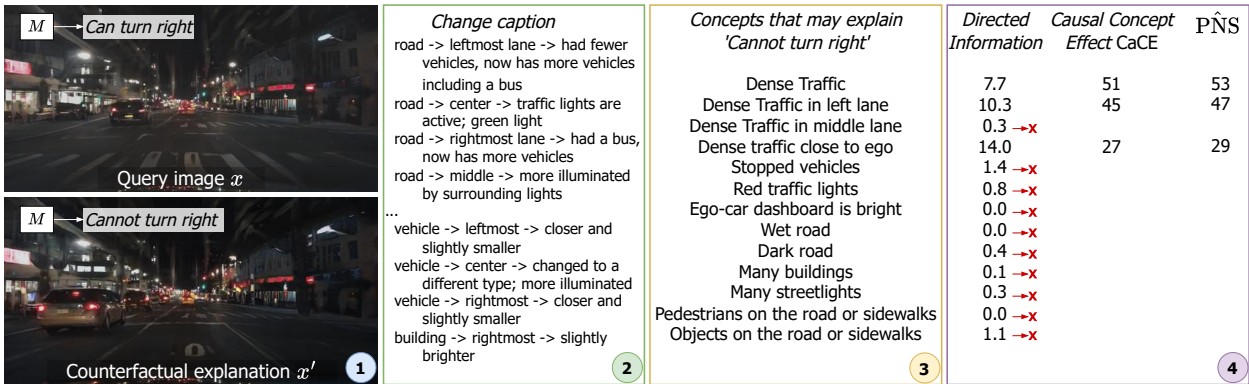

Figure 4: **GIFT output for the biased classifier on BDD-OIA (Xu et al., 2020)**. The model $M$ classifying images into '*Can/Cannot turn right*' is intentionally biased for vehicles in the left lane to yield 'Cannot turn right'. We illustrate the output of Stages 1 and 2 for a single randomly selected input and the global output of Stages 3 and 4. Causal metrics are used when DI $\geq 5\%$. DI, CaCE, $\widehat{PNS}$ in %.

right' action introduced by Zemni et al. (2023). The classifier was intentionally trained with a dataset bias, associating images with vehicles on the *left* side to the 'cannot turn *right*' class, in addition to their normal labels. The setup aims to test whether GIFT can reveal this injected bias in the model's decision-making.

**Instantiation of GIFT.** The instantiation follows the setup described in Sec. 4.2, building on general-purpose models. For counterfactual generation (Stage 1), we use OCTET (Zemni et al., 2023), which is better-suited for complex driving scenes and to enable fair comparison to baselines. Details in App. H.

**Results.** Fig. 4 shows results for each stage of GIFT, with (1) one sample pair of counterfactual images (of the many generated), (2) the change-caption extracted for it, (3) the relevant concepts that the LLM obtains over many change captions, and (4) the correlation and causal metrics computed for verification.

In the sample shown in the figure, the counterfactual image presents several changes, including a new vehicle in the left lane. Due to the complexity of the scenes, individual change captions in this dataset tend to be long and noisy. Again, Stage 3 is critical to condense a large collection of weak signals into a small list of meaningful hypotheses, which, in Stage 4, can be pre-filtered with the correlation metric (DI) and then ranked by the causal metrics. Fig. 16 (in Supp. Mat.) illustrates interventions used to confirm these rules.

We finish with three strong explanations for being unable to turn right: 'dense traffic', 'dense traffic in left-lane' (the bias!), and 'dense traffic close to ego-vehicle', with, respectively, CaCE of 51, 45, and 27% and $\widehat{PNS}$ of 53, 47 and 29%. The first and third explanations capture pertinent causes preventing the model yield 'turn right'. Importantly, the second explanation uncovered by GIFT should immediately call the attention of anyone evaluating the model.

**Comparison with State of the Art and Ablations.** Tab. 4 compares GIFT with existing explanation methods and ablations. The key insights are:

*Human inspection is insufficient (Ablation of Stages 1–3):* In the user study by Zemni et al. (2023), participants analyzed several images and corresponding classifier outputs but failed to identify the left-lane-vehicle bias. This shows that manual hypothesis generation is impractical for complex models with unsuspected biases and highlights the need for automated methods.

*Seeding from faithful explanations is critical (Ablation of Stages 1–2):* Several methods rely on LLM-generated hypotheses to detect biases (Csurka et al., 2024; Prabhu et al., 2023; D'Incà et al., 2024). However, with the biased-BDD-OIA classifier, those methods fail because the set of LLM-generated hypotheses does not contain the bias, which shows that without counterfactual guidance (Stage 1) and visual difference captioning (Stage 2), LLMs cannot uncover unforeseen biases. Details in App. H.2.

Table 4: **Left-lane vehicle bias identification on BDD-OIA.** Only GIFT successfully identifies the left-lane-vehicle bias. Methods relying on GPT4-generated, e.g., (Prabhu et al., 2023; Csurka et al., 2024; D'Incà et al., 2024) or human hypotheses (Zemni et al., 2023), fail to detect the bias. Besides, using independent captions in Stage 2, instead of contrastive ones, make the LLM miss the bias. Counterfactuals alone only help 65% of humans to detect the bias (Zemni et al., 2023).

| Explanation Method | Ablated stage | Bias Detected? |
|---|---|---|
| LLM-generated expl. e.g., Csurka et al. (2024) | 1,2 | No |
| Human-proposed expl. Zemni et al. (2023) | 1,2,3 | No |
| GIFT w/o change captions | 2 | No |
| Counterfactual expl. Zemni et al. (2023) | 2,3,4 | 65% of users |
| GIFT | - | Yes |

*Contrastive image analysis is key (Ablation of Stage 2):* In this ablation, we replace *change* captions with *independent* captions for each image in a counterfactual pair. Those independent captions do not highlight key pairwise differences, and the LLM in Stage 3 thus fails to identify the bias. This shows the importance of *change* captioning, which emphasizes the counterfactual signal and explicitly captures differences. Details in Sec. H.2 for BDD and in Sec. F.4 for CLEVR.

*Automated analysis reduces human biases (Ablation of Stages 2–4):* A simple visual inspection of Stage 1's counterfactuals led 35% of the user-study's participants to miss the bias (Zemni et al., 2023), showing that humans can overlook non-intuitive biases even when they appear clearly. GIFT automatically converts counterfactuals into explicit textual explanations with causal scores, making biases more apparent.

Overall, these findings validate the design of GIFT for uncovering non-intuitive model biases. Unlike existing methods, GIFT requires no user-defined concepts about potential biases and reveals them in plain text.

# 5 Limitations and Future Work

As GIFT builds on various models from recent literature, it inherits the limitations of its components. Below, we discuss these limitations and propose potential ways to address them in future work.

**Applicability to new domains.** The applicability of GIFT to new domains is intrinsically linked to the capabilities of its underlying components. Our current validation focuses on binary classification in natural image domains where off-the-shelf generative models are most robust. Extending the framework to specialized fields (e.g., medical imaging) may require domain-adapted models at three key stages.

• The scope of counterfactual explanations may be constrained by the generative model underlying the counterfactual method. Methods like STEEX (Jacob et al., 2022) and ACE (Jeanneret et al., 2023) limit the search space and, for instance, cannot assess the importance of object positions. OCTET (Zemni et al., 2023), which uses BlobGAN (Epstein et al., 2022), offers a partial solution by enabling disentangled representations of scene layout and semantics ; this motivates our choice of OCTET for object-centric datasets (CLEVR in Sec. 4.1 and BDD in Sec. 4.3). To mitigate the potentially limiting scope, combining multiple counterfactual methods could improve the diversity and completeness of the explanations. Despite this, our results show that grounding the explanations in local counterfactuals reveals unexpected biases and insights that human- or LLM-generated hypotheses might miss.

• Image change captioning in complex domains (Stage 2) relies on an image change captioning model, which face challenges in complex domains, such as driving scenes. These models often lack the training to interpret domain-specific or fine-grained changes, such as precise object positions or subtle attributes, which require an advanced understanding of compositional scene structure (Ma et al., 2023; Ray et al., 2023). A potential solution is to generate synthetic perturbations (e.g., with models like InstructPix2Pix (Brooks et al., 2023)) that can be used to fined-tune VLMs for more accurate change captioning.

• Image editing (Stage 4) can produce unrealistic outputs, particularly for complex scenes or compositional queries like "Shift the red car on the right of the black truck to left" (Ma et al., 2023). Consequently, the

faithfulness of our verification stage is bounded by the performance of these intervention tools: if the editor fails to insert a concept correctly (e.g., fails to add 'glasses'), the classifier will likely not change its decision, leading to a low causal CaCe score. This results in a false negative where a valid rule is rejected, reflecting a conservative design choice to prioritize precision over recall. Conversely, the introduction of confounding features (false positives) is mitigated by employing highly specific text prompts designed to isolate the causal effect of the target concept during editing. While recent advances in compositional understanding (Esser et al., 2024) show promise, further progress is needed to handle such detailed queries reliably.

**Reasoning capabilities of the LLM (Stage 3).** Stage 3 leverages an LLM to identify global explanations from noisy change captions. This requires the LLM to disambiguate local evidence and infer common explanations across multiple counterfactual changes (as explained in Sec. 3.3). Although our experiments show that LLMs perform reasonably well in this task, it remains an unconventional use case for the LLM. Future work could explore fine-tuning LLMs specifically for this task to improve their ability to reason over aggregated counterfactual signals and handle more complex scenarios. Moreover, achieving high recall for hypotheses at this stage is crucial, as errors can be addressed during Stage 4. Accordingly, prompt-engineering can be used to encourage the LLM to generate a large amount of hypotheses.

**Automated exploration of the explanation space.** After Stage 4, GIFT allows users to explore and combine concepts with observed correlation or causal signals or to test their own hypotheses. While this human-in-the-loop approach is desirable as it implies end-users in the process, fully automating the exploration process could enhance scalability. One promising direction involves using the Stage 3 LLM to iteratively reason over counterfactual descriptions while incorporating feedback from correlation (DI) and causal measurements (CaCE). This could follow the 'visual programming' paradigm (Gupta & Kembhavi, 2023), where the LLM utilizes tools like Stage 4 verification in a chain-of-thought manner (Wei et al., 2022). However, our initial experiments in this direction were unsatisfactory, highlighting the current limitations of LLM reasoning for such ambitious goals.

**Pipeline complexity.** The computational cost of GIFT (see App. I) is dominated by Stage 1, where counterfactuals are generated via per-image optimization. The remaining stages—zero-shot change captioning, hypothesis induction, and causal verification—require only forward passes and are comparatively lightweight. Computing causal metrics adds overhead due to the many image edits, but remains less costly than counterfactual generation.

Runtime was not our primary focus, as GIFT targets high-quality, testable global explanations rather than fast local attributions. Its more ambitious goals naturally entail higher cost than traditional saliency-based methods. Nonetheless, we find the trade-off acceptable: generating the hypothesis set takes about 13 minutes, with verification adding 6 minutes per hypothesis. This remains efficient relative to approaches such as Zemni et al. (2023), which rely on trained human evaluators to identify biases less systematically (see Sec. 4.3).

Regarding scalability to multi-class settings, the computational cost scales linearly with the number of *relationships* one wishes to analyze, rather than the total number of classes. GIFT is designed to explain specific decision behaviors (e.g., Why Class A and not Class B?" or Why Class A and not the rest?") rather than processing all classes simultaneously. While our experiments focus on binary classification to isolate specific decision boundaries, in a multi-class setting, the framework operates in a one-versus-rest or one-versus-one manner. Consequently, explaining a specific transition (e.g., the Top-1 prediction against the Runner-up) requires only a single run of the pipeline, making the approach viable without modification for classifiers with large label spaces.

## 6 Conclusion

GIFT offers a conceptual and grounded approach for generating global explanations of deep vision models by systematically building from local insights. Starting with instance-level counterfactual explanations, GIFT translates those findings to natural language and aggregates them to identify recurring patterns, allowing global explanations to emerge. Those explanations are then doubly checked, with VQA for high correlation and image interventions for causal effect on the model decisions, supporting the faithfulness of identified patterns to the model's decision-making. A key strength of GIFT is its flexibility. By validating the

framework with diverse models at each stage, we demonstrate its robustness on natural image domains and several use-cases, while acknowledging that its immediate applicability to specialized fields remains bounded by the capabilities of the underlying generative components. In future work, we will explore automatic hypothesis refinement, for example by feeding back causal scores to the LLM to form an iterative loop. This would preserve GIFT's causal verification rigor scaling to even more complex vision models. Besides, leveraging 'reasoning' inference abilities of LLMs (Snell et al., 2024) offers a promising direction to refine Stage 3.

## Acknowledgments

This work was partially supported by the ANR MultiTrans project (ANR-21-CE23-0032).

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

## A   Broader impact statement

By enhancing the understanding of deep learning models, particularly in vision classification, this work contributes to the responsible deployment of AI systems in safety-critical applications. While the potential societal consequences of this work are generally aligned with the positive progression of machine learning, we acknowledge that enhanced interpretability could be used in ways that are not constructive, such as to identify and exploit weaknesses in deployed systems. However, we believe that the overall impact of this research will be to improve the fairness and trustworthiness of AI systems. We will release our code and data to promote transparency and collaboration in this field, which we hope will mitigate any potential risks and will allow further research into the ethical considerations arising from model interpretation. We encourage other researchers to use and extend the GIFT framework in ways that lead to more reliable and trustworthy applications of AI.

## B   Further discussion on related works

Some prior work produces global and sometimes textual explanations. However, these approaches differ fundamentally from GIFT in both scope and capabilities, beyond the faithful (causal) criterion. Tab. 5 summarizes key differences.

Classical concept-based methods such as TCAV Kim et al. (2018) require *manually predefined concepts*. This is a strong limitation because they can only evaluate what users anticipate. As a result, these methods may miss unexpected model biases. The OCTET Zemni et al. (2023) user study shows this clearly (Sec. 4.3): no human anticipated the BDD bias even after inspecting input images and model predictions. In contrast, GIFT automatically discovers such unexpected factors without requiring any predefined concepts.

Unsupervised concept discovery methods such as ACE Ghorbani et al. (2019) and CRAFT Fel et al. (2023b) avoid predefined concepts but are architecture-specific (CNN-only) and produce concept vectors that require manual interpretation through visualization (e.g., (Fel et al., 2023a)) or prototype inspection. This makes them unsuitable for holistic, model-agnostic explainability. GIFT, instead, directly outputs interpretable natural language explanations and applies to any differentiable classifier, providing causal verification of global explanations.

## C   Details on hypotheses verification (Stage 4)

### C.1   Directed information (DI)

We provide detailed steps for computing Directed Information (DI), described in Sec. 3.4. It evaluates the correlation between concepts $c_e$ and class label $y$. DI is computed as:

$$\mathrm{DI}(c_e, y) = \frac{I(c_e; y)}{H(c_e)},$$

where $I(c_e; y)$ is the mutual information and $H(c_e)$ is the entropy. Below, we explain each term and outline how they are computed.

**Mutual Information,** $I(c_e; y)$   Mutual information quantifies the amount of information the presence of concept $c_e$ provides about the predicted class label $y$. It is defined as:

$$I(c_e; y) = \sum_{c_e, y} p(c_e, y) \log \left( \frac{p(c_e, y)}{p(c_e)p(y)} \right),$$

where:

- $p(c_e, y)$ is the joint probability of the concept $c_e$ and class label $y$,

Table 5: Comparison of global explainability methods. GIFT differs by producing faithful (causal) textual explanations without requiring predefined concepts, human interpretation, or architecture-specific components.

| Method | Scope | Interpretability | Predefined Concepts | Interpretation Needed | Architecture Specific | Faithfulness |
|---|---|---|---|---|---|---|
| **GIFT** | Global | High (text) | No | No | No | Yes (causality) |
| Classical concept-based (e.g., TCAV Kim et al. (2018)) | Global | High (text) | Yes (manually defined) | No | No | No (correlations) |
| Unsupervised concept-based (e.g., ACE Ghorbani et al. (2019), CRAFT Fel et al. (2023b)) | Global | Low (vectors) | No | Yes (manual visualization) | Yes (CNN-specific) | No (correlations) |

- $p(c_e)$ is the marginal probability of $c_e$,

- $p(y)$ is the marginal probability of $y$.

We use the outputs of a Visual Question Answering (VQA) model to estimate $p(c_e, y)$. The VQA model predicts whether the concept $c_e$ is present in an input image $x$. For a dataset of images $\mathcal{X}$, we calculate:

$$p(c_e, y) = \frac{1}{|\mathcal{X}|} \sum_{x \in \mathcal{X}} \mathbb{1}[\text{VQA}(x, c_e) = 1 \land M(x) = y],$$

where $\mathbb{1}[\cdot]$ is the indicator function, and $|\mathcal{X}|$ the cardinal of the set $\mathcal{X}$. The marginal $p(c_e)$ and $p(y)$ are derived from the joint probabilities:

$$p(c_e) = \sum_y p(c_e, y), \quad p(y) = \sum_{c_e} p(c_e, y).$$

**Entropy of the Concept Presence,** $H(c_e)$   Entropy measures the uncertainty in the presence of concept $c_e$ across the dataset. It is defined as:

$$H(c_e) = -\sum_{c_e} p(c_e) \log p(c_e).$$

We use the marginal probability $p(c_e)$ estimated above.

### C.2   Probability of Necessary and Sufficient Cause (PNS)

We consider $Y$, a random variable that encodes the values of the label of a data point $x \in \mathcal{X}$. Following Tian & Pearl (2000), Wang & Jordan (2021) define, for a given class value $y$ and data representation $Z$, the probability of the given representation to be a necessary cause ($\text{PN}_{Z,y}$) and to be a sufficient cause ($\text{PS}_{Z,y}$) of $Y = y$. We use a similar theoretical framework in this study but use the notion of concept $c_e$, being present ($c_e = 1$) or absent ($c_e = 0$) in the data point, instead of a representation. The formula for $\text{PN}_{c_e,y}$ and its empirical estimate $\widehat{\text{PN}}_{c_e,y}$ are given below:

$$\text{PN}_{c_e,y} = \mathbb{P}(M(x^{-c_e}) \neq y | M(x) = y, x \in \mathcal{X}_{c_e=1}) \tag{5}$$

Each one of the descriptions below (separated by '—') details the changes a pair of images underwent as part of a counterfactual analysis for a machine-learning binary classifier. The described changes caused the classifier to change its prediction either from class 0 to class 1 or from class 1 to class 0. Your task is to consider ALL the descriptions and summarize the main factors leading the classifier to choose class 0 or 1.
- The counterfactual analysis is noisy, and the descriptions may contain irrelevant or contradictory information. Your task is to focus on the most important factors that appear consistently across many instances
- Find testable factors that can be observed and measured in the images (e.g., objects' presence, appearance, arrangement, etc.)
- Present your factors as a bulleted list
—

From class 0 to class 1:
left car → closer; more visible headlights
ego-car → dashboard → brighter
. . .
—

From class 0 to class 1:
traffic → leftmost lane → was clear, now a taxi is present
traffic → rightmost lane → was clear, now is occupied by cars
. . .

Figure 5: **Prompt used for the LLM in Stage 3.** This prompt is utilized across all experimental use cases. The ellipses '. . .' are placeholders, replaced with the concatenation of all change captions gathered in stage 2. Note that the class labels are not provided; instead, they are represented generically as 0 or 1.

$$\widehat{\text{PN}}_{c_e,y} = \begin{cases} \dfrac{\sum\limits_{x \in \mathcal{X}_{c_e=1}} \mathbb{1}[M(x^{-c_e}) \neq y \wedge M(x)=y]}{\sum\limits_{x \in \mathcal{X}_{c_e=1}} \mathbb{1}[M(x)=y]} & \text{if } \sum\limits_{x \in \mathcal{X}_{c_e=1}} \mathbb{1}[M(x)=y] \neq 0 \\ 0 & \text{otherwise.} \end{cases} \tag{6}$$

Similarly, for $\text{PS}_{c_e,y}$ and its empirical estimate $\widehat{\text{PS}}_{c_e,y}$:

$$\text{PS}_{c_e,y} = \mathbb{P}(M(x^{+c_e}) = y | M(x) \neq y, x \in \mathcal{X}_{c_e=0}) \tag{7}$$

$$\widehat{\text{PS}}_{c_e,y} = \begin{cases} \dfrac{\sum\limits_{x \in \mathcal{X}_{c_e=0}} \mathbb{1}[M(x^{+c_e})=y \wedge M(x) \neq y]}{\sum\limits_{x \in \mathcal{X}_{c_e=0}} \mathbb{1}[M(x) \neq y]} & \text{if } \sum\limits_{x \in \mathcal{X}_{c_e=0}} \mathbb{1}[M(x) \neq y] \neq 0 \\ 0 & \text{otherwise.} \end{cases} \tag{8}$$

One can then define the probability $\text{PNS}_{c_e,y}$ of $c_e$ being both a necessary and a sufficient cause of $Y = y$, as the following weighted combination of $\text{PN}_{c_e,y}$ and $\text{PS}_{c_e,y}$:

$$\text{PNS}_{c_e,y} = \mathbb{P}(M(x) = y, x \in \mathcal{X}_{c_e=1}) \cdot \text{PN}_{c_e,y} + \mathbb{P}(M(x) \neq y, x \in \mathcal{X}_{c_e=0}) \cdot \text{PS}_{c_e,y}. \tag{9}$$

We approximate $\text{PNS}_{c_e,y}$, using again empirical estimates, to obtain our metric (Eq. 4):

$$\begin{aligned} \widehat{\text{PNS}}_{c_e,y} &= \frac{\sum\limits_{x \in \mathcal{X}_{c_e=1}} \mathbb{1}[M(x) = y]}{|\mathcal{X}|} \cdot \widehat{\text{PN}}_{c_e,y} + \frac{\sum\limits_{x \in \mathcal{X}_{c_e=0}} \mathbb{1}[M(x) \neq y]}{|\mathcal{X}|} \cdot \widehat{\text{PS}}_{c_e,y} \\ &= \frac{1}{|\mathcal{X}|} \left( \sum\limits_{x \in \mathcal{X}_{c_e=1}} \mathbb{1}[M(x^{-c_e}) \neq y \wedge M(x) = y] + \sum\limits_{x \in \mathcal{X}_{c_e=0}} \mathbb{1}[M(x^{+c_e}) = y \wedge M(x) \neq y] \right). \end{aligned} \tag{10}$$

A first quality of $\widehat{\text{PNS}}_{c_e,y}$ as a metric is that, contrarily to $\text{CaCE}_{c_e}$, it is not only specific to a given concept but also to a given class. Thus, it can be freely extended to settings beyond binary and exclusive classes, while still being interpretable.

Furthermore, $\text{PNS}_{c_e,y}$ can be lower bounded using the difference of two intervention distributions (Tian & Pearl, 2000). This bound, given below, can also be used to get further causal signal evidence.

$$\text{PNS}_{c_e,y} \geq \mathbb{P}(M(x) = y | \text{do}(x \in \mathcal{X}_{c_e=1})) - \mathbb{P}(M(x) = y | \text{do}(x \in \mathcal{X}_{c_e=0})). \tag{11}$$

### C.3 Relation between $\text{CaCE}_{c_e}$ and $\widehat{\text{PNS}}_{c_e,y}$

Let us prove that for $\{0,1\}$-class settings, we have $\text{CaCE}_{c_e} = \widehat{\text{PNS}}_{c_e,y=1} - \widehat{\text{PNS}}_{c_e,y=0}$. We use Eq. 3 that gives us $\text{CaCE}_{c_e}$ (Goyal et al., 2019a) as follows:

$$\text{CaCE}_{c_e} = \frac{1}{|\mathcal{X}|} \left( \sum_{x \in \mathcal{X}_{c_e=1}} M(x) - M(x^{-c_e}) + \sum_{x \in \mathcal{X}_{c_e=0}} M(x^{+c_e}) - M(x) \right). \tag{12}$$

Using the $\{0,1\}$-class setting, we can rewrite $\text{CaCE}_{c_e}$'s equation using the indicator function ($\mathbb{1}[\cdot]$):

$$
\begin{aligned}
\text{CaCE}_{c_e} =& \frac{1}{|\mathcal{X}|} \Bigg( \sum_{x \in \mathcal{X}_{c_e=1}} (\mathbb{1}[M(x) = 1 \wedge M(x^{-c_e}) = 0] - \mathbb{1}[M(x) = 0 \wedge M(x^{-c_e}) = 1]) \\
&+ \sum_{x \in \mathcal{X}_{c_e=0}} (\mathbb{1}[M(x^{+c_e}) = 1 \wedge M(x) = 0] - \mathbb{1}[M(x^{+c_e}) = 0 \wedge M(x) = 1]) \Bigg) \\
=& \frac{1}{|\mathcal{X}|} \Bigg( \sum_{x \in \mathcal{X}_{c_e=1}} \mathbb{1}[M(x) = 1 \wedge M(x^{-c_e}) = 0] + \sum_{x \in \mathcal{X}_{c_e=0}} \mathbb{1}[M(x^{+c_e}) = 1 \wedge M(x) = 0] \Bigg) \\
&- \frac{1}{|\mathcal{X}|} \Bigg( \sum_{x \in \mathcal{X}_{c_e=1}} \mathbb{1}[M(x) = 0 \wedge M(x^{-c_e}) = 1] + \sum_{x \in \mathcal{X}_{c_e=0}} \mathbb{1}[M(x^{+c_e}) = 0 \wedge M(x) = 1] \Bigg) \\
=& \widehat{\text{PNS}}_{c_e,y=1} - \widehat{\text{PNS}}_{c_e,y=0}.
\end{aligned}
\tag{13}
$$

Eq. 13 shows that $\text{CaCE}_{c_e} = \widehat{\text{PNS}}_{c_e,y=1}$ (resp. $\text{CaCE}_{c_e} = \widehat{\text{PNS}}_{c_e,y=0}$) if and only if $\widehat{\text{PNS}}_{c_e,y=0} = 0$ (resp. $\widehat{\text{PNS}}_{c_e,y=1} = 0$), i.e., if and only if the causal effect is purely unidirectional with $c_e$ causing $y = 1$ and never $y = 0$ (resp. $y = 0$ and never $y = 1$).

## D  LLM prompts (Stage 3).

**Default LLM prompt for Stage 3.** We report in Fig. 5 the prompt that we use to summarize change captions and make global hypotheses emerge. The prompt is exemplified for the BDD experiment but is identical for CLEVR and CelebA experiments.

**LLM prompt (Stage 3) for the ablation of Stage 2.** In the baseline, described in Sec. 4.3 ('Ablation of Stage 2'), instead of feeding the LLM with *change captions*, we provide simple captions for all images and counterfactuals. To generate the captions for each image, we use the recent Florence-2 (Xiao et al., 2024), prompted with the 'More detailed caption' task mode. Then, we feed all the captions, along with the classification yielded by the target model $M$, to the LLM, with a prompt shown in Fig. 14. LLM fails to hypothesize that Class '1' is confounded by the presence of vehicles in the left-lane. This ablation highlights the importance of fine-grained pairwise comparisons between an image and its counterfactual explanation, as provided by Stage 2. They are crucial for the effectiveness of our method.

Images classified in class "0" by the model $M$        Images classified in class "1" by the model $M$

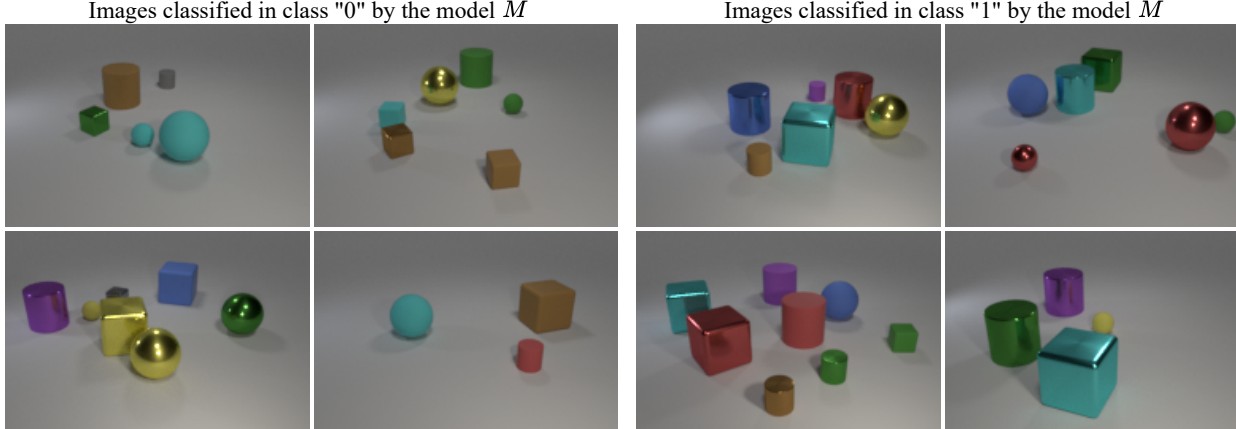

Figure 6: **Can you find the class meaning?** We propose a challenge to the reader. The classifier $M$ has been trained to recognize a specific visual rule, e.g., 'images in class 1 contain a yellow rubber object', as with the experiments of Sec. 4.1. Can you guess what is the rule simply by observing the images and their classification given by the model $M$? This challenge illustrates the difficulty of the CLEVR data domain: CLEVR images hide complex compositional potential behind their minimalist appearance, with diverse colors, textures, and shapes. The answer is given in the footnote [1].

**LLM prompt (Stage 3) for the ablation of Stages 1 and 2.** In the baseline, described in Sec. 4.3 ('Ablation of Stages 1 and 2'), we prompt the LLM of Stage 3 to imagine possible biases that the target classifier may have. There are thus no counterfactual explanations generation and the LLM does not depend on any information given by the classifier, except for the class meaning (=labels). We show the prompt and output in Fig. 15. As there are no dependencies anymore on the target classifier, the LLM can only imagine generic biases and fails to find the unexpected left-lane bias. As LLM-generated hypotheses fail to raise the left-lane-vehicle bias, it shows the importance of the counterfactual guidance (Stages 1 and 2) in the GIFT framework.

# E VLM prompts (Stage 2)

We report in Fig. 12 and Fig. 13 the prompts that we use to prompt Pixtral for image change captioning, for BDD and CelebA experiments respectively. We use a one-shot in-prompt annotated sample as guidance, where we manually describe the differences between an image (randomly selected) and the obtained counterfactual explanation after Stage 1. We qualitatively find that this guides the VLM to produce more accurate change captions.

In the case of CLEVR, we use CLIP4IDC which is trained for the image change captioning task on the CLEVR domain, with the annotated CLEVR-Change (Park et al., 2019) dataset.

# F Experimental Details on CLEVR

## F.1 CLEVR: a challenging compositional domain

Fig. 6 presents a challenge to the reader to uncover the underlying visual rule learned by the classifier $M$. In this example, $M$ has been trained on a specific rule, such as 'images in class 1 contain a yellow rubber object' or 'images in class 0 contain a red object', similar to the setup described in Sec. 4.1. By observing the images and the model's predictions, the reader is invited to deduce the rule. This task highlights the inherent complexity of the CLEVR dataset, where simple appearances mask intricate combinations of colors, textures, and shapes. The solution to the challenge is provided in the footnote at the end of the figure caption.

## F.2 CLEVR Classifiers

We use CLEVR-Hans Stammer et al. (2021) to generate six binary classification datasets with BLENDER. The images are then resized and center-cropped to 128×128 to match the resolution of the BlobGAN generative model. The training and validation sets contain 3,000 and 300 samples, respectively, with balanced labels. We train 12 classifiers (listed in Tab. 2) with either ViT or ResNet34 backbones on these datasets, selecting the checkpoint with the best validation accuracy from the first three epochs. For all these classifiers, we report near-perfect validation accuracies, ranging from 96.3% to 99.3%.

## F.3 GIFT Instantiation

**Stage 1.** To instantiate GIFT on the CLEVR domain, we first require a method for counterfactual explanation (Stage 1). For this, we employ OCTET (Zemni et al., 2023), which produces 'object-aware' explanations and is well-suited for the object-centric nature of CLEVR. However, since the official OCTET implementation is not compatible with CLEVR, we reimplemented it for this domain. A key component of this adaptation is BlobGAN (Epstein et al., 2022), a compositional generative model capable of editing, inserting, and removing objects in a differentiable manner. We trained BlobGAN on the original CLEVR dataset. The images were resized and center-cropped to 128×128 resolution. We used $K = 15$ blobs, slightly exceeding the maximum number of objects (10) in a CLEVR scene. The BlobGAN training was conducted for 181 epochs. Notably, we did not need to retrain an image encoder. Instead, we directly sample within BlobGAN's latent space to generate the original query image. This approach not only produced satisfying results but also circumvented the need for computationally intensive image reconstructions, which are typically required in the first step of OCTET optimization.

**Stage 2.** For the image change captioning (Stage 2), we use CLIP4IDC (Guo et al., 2022), which is specifically designed for the CLEVR dataset. This model is trained on CLEVR-Change (Park et al., 2019), a dataset that includes image pairs and textual *change captions* describing the differences between the images. We note that since the CLEVR-Change dataset does include shape or size changes (e.g., a sphere turning into a cylinder, or a small sphere is now a big sphere), CLIP4IDC is unable to describe such transformations. As a result, we only consider rules that do not rely on shape or size, as counterfactual explanations produced with OCTET would include such changes that CLIP4IDC cannot translate into natural language. In contrast, for the other use-cases with CelebA and BDD-OIA, we do not face this limitation as we use generalist VLMs. However, these models are somewhat noisier since they were not specifically trained for this task.

**Stage 4.** In Stage 4, hypothesis verification was conducted using 200 images: 100 classified as 1 by the model and 100 classified as 0. Importantly, we never accessed the ground-truth labels. For concept verification in Stage 4 and the DI computation, we utilized the MiniCPM (Yao et al., 2024) model. Interventions in Stage 4 involved object additions and removals:

- Object addition: We identified empty regions in the image by checking if the mean pixel value and standard deviation within a selected area were close to the background color and zero, respectively. This produced a mask that was passed to Stable Diffusion (Esser et al., 2024) alongside the description of the object to add, e.g., 'a cyan object'. The model then placed the object in the specified region.

- Object removal: We identified the bounding boxes of the objects to remove using Florence-2 (Xiao et al., 2024) in 'CAPTION_TO_PHRASE_GROUNDING' mode. Subsequently, Stable Diffusion (Esser et al., 2024) was used to inpaint these regions with the word 'background', effectively removing the objects.

---

[1] Answer: 'Images of class 1 contain a cyan metallic object.'

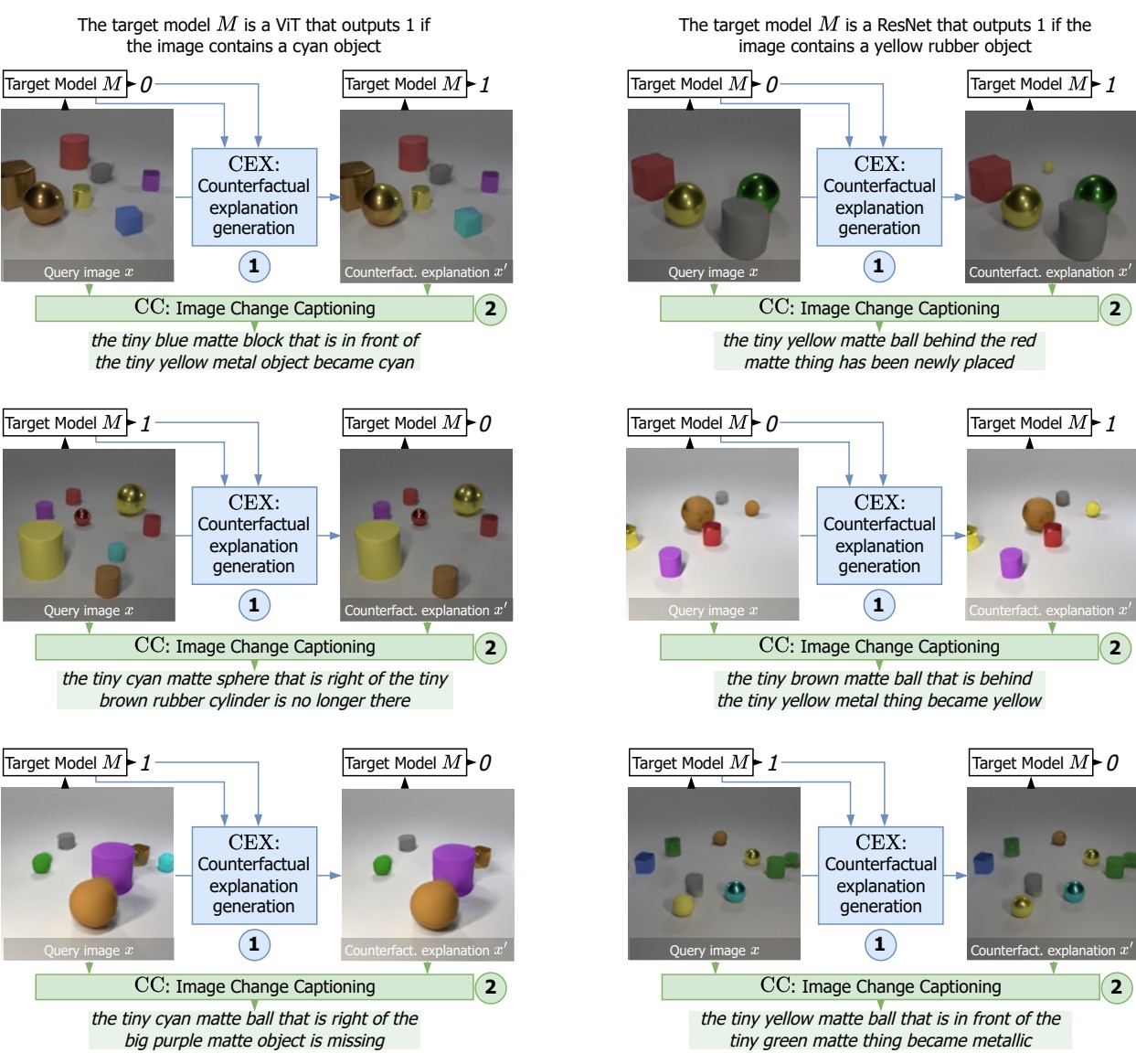

Figure 7: **Visualization of the Stages 1 and 2** for classifiers trained on CLEVR data. CEX generates a counterfactual explanation and CC describes differences in plain text. Examples on the left block are obtained for a model $M$ that outputs '1' if the image contains a cyan object. Examples on the right block are obtained for a model $M$ that outputs '1' is the image contains a yellow rubber object. The counterfactual explanations exhibit notable diversity, encompassing changes in object colors, appearances, disappearances, and textures. Furthermore, the CLIP4IDC model, specifically trained for this task, excels at translating visual changes into natural language descriptions. Its outputs are consistently accurate, with minimal noise and rare errors.

Table 6: **Complete table of experiments run in Sec. 4.1.** The explanations that obtained the highest scores for each classifier and metric are shown in bold. In all cases except one, the rule used to train the target classifier was proposed by the LLM after Stage 3. The rule always corresponds to the hypothesis with the highest CaCE and $\widehat{\text{PNS}}$ measurement. The use of italics for 'red metal object' indicates that the rule was manually created by recombining the first two rules found in Stage 3.

| Rule (Ground Truth) | Architecture | Output of Stage 3 Explanation: 'Class 1 contains a...' | Output of Stage 4 DI (%) | CaCE (%) | $\widehat{\text{PNS}}$ (%) |
|---|---|---|---|---|---|
| cyan object | ResNet34 | **cyan object** | **96.8** | **62.2** | **62.2** |
| | | cyan metal object | 26.0 | 50.9 | 50.9 |
| | | cyan rubber object | 31.2 | 47.6 | 47.6 |
| purple object | ResNet34 | **purple object** | **92.9** | **71.2** | **71.2** |
| metal object | ResNet34 | red object | 0.2 | 8.6 | 9.1 |
| | | **metal object** | **88.9** | **24.2** | **24.2** |
| rubber object | ResNet34 | **rubber object** | **66.5** | **29.8** | **29.8** |
| yellow rubber object | ResNet34 | **yellow rubber object** | **44.3** | **67.2** | **67.7** |
| | | small yellow rubber sphere | 14.1 | 57.6 | 57.6 |
| | | yellow object | 31.0 | 38.4 | 39.9 |
| red metal object | ResNet34 | red object | 46.8 | 42.9 | 43.9 |
| | | metal object | 3.6 | 11.6 | 13.6 |
| | | *red metal object* | *65.5* | *61.6* | *61.6* |
| cyan object | ViT | **cyan object** | **92.9** | **63.1** | **63.1** |
| | | cyan sphere | 24.4 | 50.5 | 50.5 |
| | | cyan cube | 28.0 | 42.4 | 43.4 |
| purple object | ViT | purple sphere | 17.2 | 52.5 | 52.5 |
| | | purple cylinder | 29.6 | 53.0 | 53.0 |
| | | **purple object** | **92.9** | **70.2** | **70.2** |
| metal object | ViT | **metal object** | **83.4** | **37.4** | **37.4** |
| | | rubber object | 16.3 | 5.6 | 6.6 |
| | | red metal object | 14.7 | 36.4 | 36.9 |
| | | yellow metal object | 14.7 | 34.8 | 35.4 |
| | | green metal object | 10.8 | 29.3 | 29.3 |
| rubber object | ViT | **rubber object** | **62.0** | **30.8** | **31.3** |
| | | blue object | 0.0 | 8.6 | 9.6 |
| | | cyan object | 0.6 | 7.6 | 8.1 |
| yellow rubber object | ViT | **yellow rubber object** | **51.2** | **71.7** | **71.7** |
| | | yellow object | 42.4 | 49.0 | 50.0 |
| red metal object | ViT | metal object | 0.3 | 12.6 | 14.1 |
| | | **red metal object** | **63.2** | **70.7** | **70.7** |
| | | red object | 40.1 | 43.4 | 43.9 |
| | | rubber object | 1.3 | 8.6 | 10.6 |
| | | cube | 0.2 | 11.1 | 13.6 |
| | | sphere | 0.0 | 9.6 | 11.1 |

## F.4   Results

**Qualitative samples after Stages 1 and 2.**   Fig. 7 provides qualitative examples from Stages 1 and 2, highlighting the outputs of the counterfactual generator (CEX) and the change captioner (CC). The

counterfactual explanation optimization depends on three inputs: the original image, the model decision to be flipped, and the model itself (illustrated with blue boxes and arrows). In contrast, the image change captioning stage relies solely on the pair of input images (indicated by the green box). The counterfactual explanations exhibit notable diversity, encompassing changes in object colors, appearances, disappearances, and textures. Furthermore, the CLIP4IDC model, specifically trained for this task, excels at translating visual changes into natural language descriptions. Its outputs are consistently accurate, with minimal noise and rare errors.

**Complete results.** Tab. 6 summarizes the results of Stages 3 and 4 across the 12 classifiers tested on the CLEVR dataset. Notably, in all but one case, the rule that governed the target classifier was correctly proposed by the large language model (LLM) after Stage 3. Stage 3 outputs reveal that the LLM generates a small set of hypotheses (between 1 and 5). In certain instances, only a single hypothesis is produced, which happens to be correct. Still, achieving high recall for hypotheses at this stage is crucial, as errors can be addressed during Stage 4. Importantly, the rule always corresponds to the hypothesis with the highest CaCE measurement, ensuring a robust and systematic selection process.

**Ablation of Stage 2.** Tab. 7 presents the results of Stage 3 for the experiment in which we ablate the change captioning component (Stage 2). In this experiment, the *change captions* are replaced with independently acquired *captions* for the images and their counterfactuals, using a prompt provided in Fig. 14. This modification significantly reduces performance: the rule is correctly identified for only 4 out of the 12 classifiers tested, as shown in Tab. 2. These results highlight the critical importance of fine-grained pairwise comparisons between an image and its counterfactual explanation. Such comparisons, facilitated by change captioning, are essential for the success of our method.

**Stage 3 adaptability to the captions' information.** In our main experiment, we used a large number of counterfactuals, approximately $N = 100$ image pairs, as reported in the paper. This was our initial attempt, designed to ensure fairness without tuning the number of pairs. In the ablation study, we find that as few as three pairs ($N = 3$), and occasionally even two, are sufficient for the LLM to hypothesize the correct rule. This is because the change captioning stage (Stage 2) is minimally noisy, as it uses CLIP4IDC, which is specifically trained for the image change captioning task. Additionally, the simplicity of the rule further reduces the need for a large number of counterfactual explanations.

**Stage 4 importance for selecting the best rules.** Stage 4 is critical to select and rank the hypotheses after Stage 3. As mentioned in Sec. 4.1, each trial on CLEVR raised 2 to 6 viable explanations, which Stage 4 then correctly ranked 11 out of 12 times (Tab. 2). In Sec. 4.2, we find dozens of candidate concepts, which Stage 4 reduces to a handful of promising ones (Tab. 3).

## G Experimental Details on CelebA

### G.1 GIFT Instantiation

The target model $M$ is a DenseNet121 (Huang et al., 2017), trained to classify face images as either 'old' or 'young,' following prior work (Jacob et al., 2022; Jeanneret et al., 2023).

For counterfactual generation (Stage 1), we leverage the diffusion-based model ACE (Jeanneret et al., 2023), using its official implementation.

In the image change captioning stage (Stage 2), we employ Pixtral-12B VLM (Agrawal et al., 2024). The prompt used for this stage is shown in Fig. 13. To guide the VLM, we include a one-shot annotated sample within the prompt, where we manually describe the differences between a randomly selected image and its corresponding counterfactual explanation from Stage 1. This guidance qualitatively improves the accuracy of the generated change captions.

Global explanation identification (Stage 3) is performed zero-shot using Qwen 2.5, a 72B parameter model, quantized to 8 bits (Yang et al., 2024). The prompt used for this stage is provided in Fig. 5.

Table 7: **Ablation of Stage 2: Complete Output.** We replace the *change captions* with simple independently acquired *captions* for images and counterfactuals, using the prompt shown in Fig. 14. The column 'Correct hypothesis' indicates whether the hypothesis corresponds to the rule used to train the classifier. The rule is found for only 4 out of the 12 classifiers tested. This ablation highlights the importance in our method of fine-grained pairwise comparisons between an image and its counterfactual explanation

| Rule | Architecture | Explanation: 'Class 1 contains a...' | Correct hypothesis |
|---|---|---|---|
| yellow rubber object | ViT | metal object | |
| metal object | ViT | metal object | ✓ |
| metal object | ResNet34 | green object
metal object
red object and yellow object
sphere and cube | ✓ |
| cyan object | ResNet34 | blue sphere
yellow or gray object | |
| cyan object | ViT | gray sphere
yellow object | |
| purple object | ViT | purple object | ✓ |
| purple object | ResNet34 | purple object
purple sphere
green object and purple object | ✓ |
| red metal object | ResNet34 | red object
yellow object
red object and yellow object | |
| red metal object | ViT | red object
metal object
red object and metal object | |
| rubber object | ResNet34 | yellow object
green object
metal object | |
| rubber object | ViT | blue object
purple object
yellow object and purple object
red object and yellow object | |

For hypothesis verification and interventions in Stage 4, we use 200 images: 100 classified as 'young' and 100 classified as 'old' by the target classifier $M$.

- The VQA model for this stage is MiniCPM (Yao et al., 2024).

- Global interventions: For concepts that require global adjustments, such as increasing lighting, we use image-conditioned Stable Diffusion 3 (Esser et al., 2024).

- Local interventions: For concepts requiring localized changes, we first identify the region to be edited using Florence-2 (Xiao et al., 2024) for open vocabulary detection. This provides the necessary masks, which are then passed to Stable Diffusion 2 (Rombach et al., 2022), along with a prompt stating to add or remove the concept, resulting in the post-intervention image.

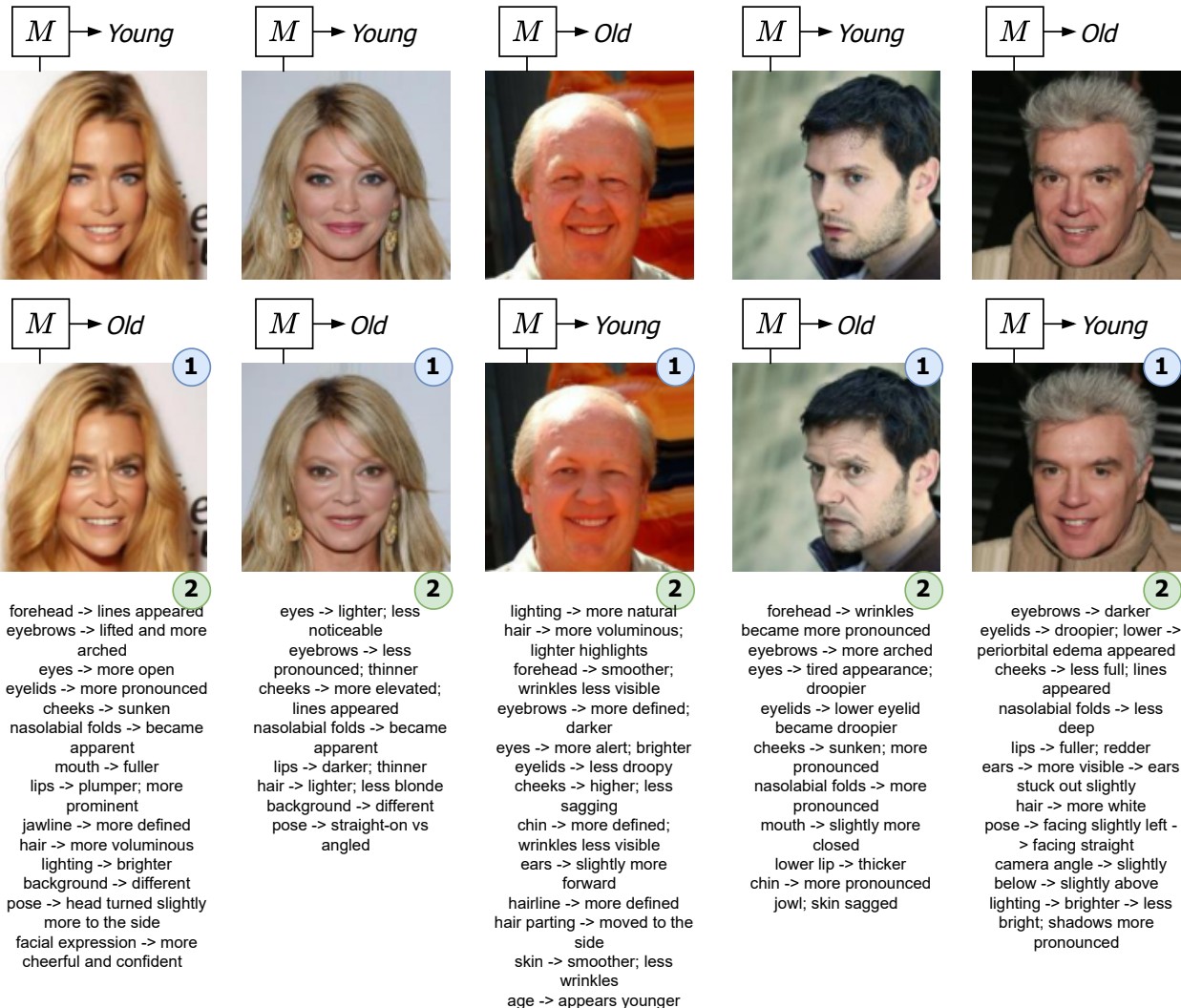

Figure 8: **Visualization of Stages 1 and 2 for the CelebA 'Young/Old' Classifier**. The top row shows the original images. The second row displays the corresponding counterfactual explanations generated in Stage 1 using the diffusion-based ACE method (Jeanneret et al., 2023). The third row presents the change captions produced in Stage 2 by the Pixtral model, which describe differences between the original images and their counterfactual explanations.

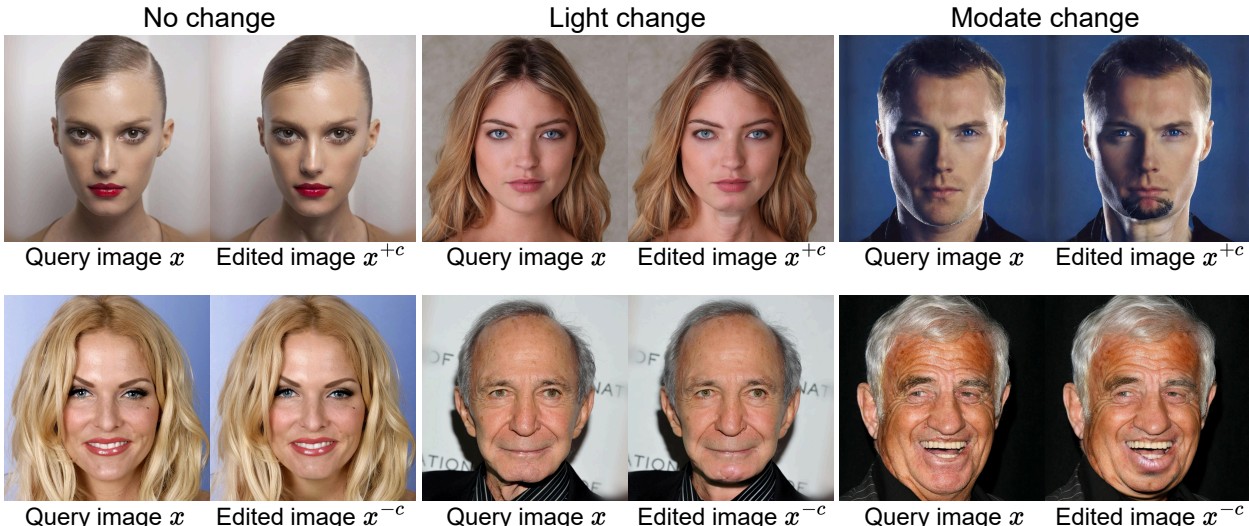

Figure 9: Examples of adding (top) or removing (bottom) *neck wrinkles*, illustrating the three manual categories: no change, light change, and moderate change.

## G.2 Results

**Qualitative samples after Stages 1 and 2.** Fig. 8 provides qualitative examples from Stages 1 and 2 in the analysis of the CelebA 'Young/Old' classifier. The counterfactual explanations generated by ACE (Jeanneret et al., 2023) highlight subtle changes, such as adding facial wrinkles or smoothing the skin. In Stage 2, the Pixtral model produces detailed change captions that describe point-by-point differences between the original images and their corresponding counterfactual explanations. While most of the caption content is accurate and meaningful, we observe occasional noise and hallucinations from the VLM, where it describes changes that are not present in the images.

**Noise from Stage 2.** We observe noticeably higher noise levels by using the zero-shot Pixtral in Stage 2, rather than the domain specific CLIP4IDC in the CLEVR experiments. To quantify this, we conducted a small diagnostic study on five randomly selected counterfactual pairs for the *Young/Old* classifier. Across these samples, the VLM produced 18 incorrect or hallucinated change descriptions out of 55 total described changes (e.g., spurious changes in pose, background, or hair volume), corresponding to an estimated hallucination rate of approximately 33%.

**Image editing model assumptions.** Our causal verification relies on the assumption that interventions introduce or remove only the target concept while leaving other attributes unchanged. This assumption is standard in causal-intervention work (e.g., Goyal et al. (2019a)). To make it hold as well as possible in practice, we design the editing procedure to be highly localized: we use spatial masks together with mask-guided text-based editing, which restricts the modification to a specific region and largely prevents unintended global changes.

To evaluate the quality of these interventions, we manually inspected edits on CelebA for three fine-grained attributes. For each concept, and for both adding and removing operations, we reviewed 50 edited images and assigned outcomes to one of three categories: *no change*, *light change*, and *moderate change*. Fig. 9 and Fig. 10 illustrate typical examples from each category for the 'neck wrinkles' and 'forehead wrinkles' attributes respectively. Tab. 8 shows the results of this analysis.

Overall, edits remain well targeted: most interventions produce either no side effects or only light ones, particularly when *removing* concepts. When secondary changes appear, they are not systematic, e.g., adding a concept may occasionally co-occur with unrelated attributes, but these side effects differ across samples

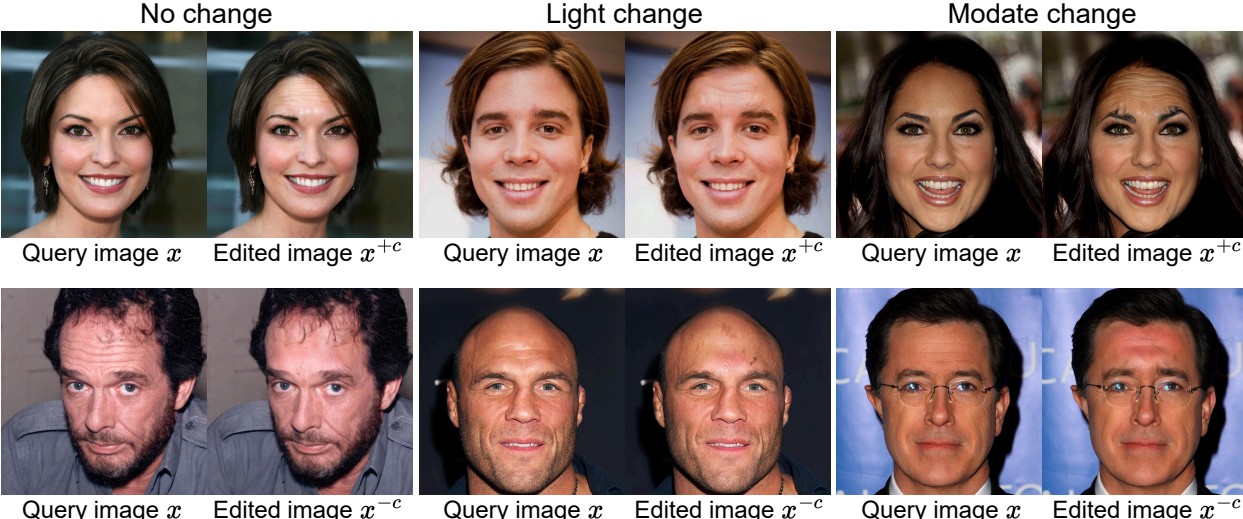

Figure 10: Examples of adding (top) or removing (bottom) *forehead wrinkles*, covering the three quality categories.

Table 8: **Intervention quality on CelebA.** Percentage of inspected samples (50 per setting) falling into each category when adding or removing a concept.

| Concept | Adding concept | | | Removing concept | | |
|---|---|---|---|---|---|---|
| | No change | Light | Moderate | No change | Light | Moderate |
| Receding hairline | 54 | 30 | 16 | 82 | 18 | 0 |
| Neck wrinkles | 52 | 24 | 24 | 52 | 26 | 22 |
| Forehead wrinkles | 48 | 30 | 22 | 68 | 22 | 10 |

and do not correlate with the manipulated concept. This lack of consistency prevents them from influencing downstream causal inference.

**Complete results.** Tab. 9 lists all the explanations associated with the class 'Old,' as generated by GIFT, along with the corresponding DI and CaCE measurements. The identified attributes include plausible ones, such as 'wrinkles on the forehead' and 'receding hairline,' as well as more unexpected ones, such as a 'detailed background' or 'low camera angle.'

# H   Experimental Details on BDD-OIA bias discovery

## H.1   GIFT Instantiation

The instantiation of GIFT for use-case 3, with the bias discovery, follows the same process as in the CelebA use-case, with the exception of counterfactual explanation generation. For this, we use OCTET (Zemni et al., 2023), which comes with the BlobGAN (Epstein et al., 2022) generative model for the BDD image domain. The biased 'Turn right' classifier used in OCTET serves as the target classifier for this use case.

## H.2   Results

**Qualitative samples after Stages 1 and 2.** Fig. 11 provides qualitative examples from Stages 1 and 2 in the analysis of the BDD-OIA 'Can/Cannot turn right' classifier. The counterfactual explanations generated by OCTET (Zemni et al., 2023) highlight sparse semantic changes, despite significant pixel value changes.

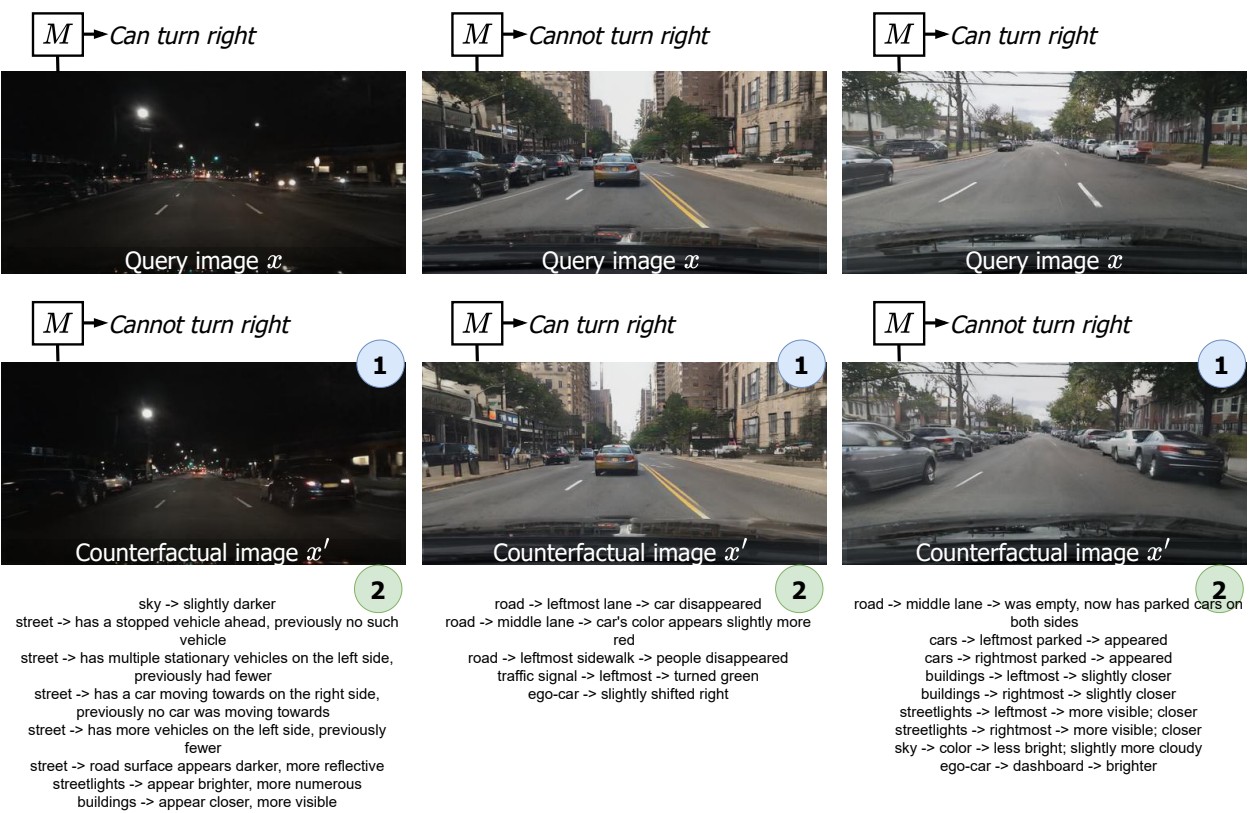

Figure 11: **Visualization of the Stages 1 and 2** for the biased turn-right classifier trained on BDD-OIA data. The top row show original images. The second row show corresponding counterfactual explanations, obtained after Stage 1 with OCTET (Zemni et al., 2023). The third row shows change captions that we obtain after Stage 2, with the Pixtral change captioning.

Table 9: **Complete output of GIFT on the CelebA 'Old' classifier.** Evaluation with causal metrics when DI ≥ 15%.

| Concepts associated to the class 'Old' | DI (%) | CaCE (%) | $\widehat{\text{PNS}}$ (%) |
|---|---|---|---|
| *after Stage 3* | | | |
| Receding Hairline | 29.9 | 2.0 | 5.0 |
| Neck Wrinkles | 28.1 | 3.0 | 3.5 |
| Wrinkles on Forehead | 26.4 | 6.5 | 7.0 |
| Wrinkles around Eyes | 20.7 | 6.5 | 7.0 |
| Glasses | 16.5 | 11.5 | 16.0 |
| Drooping Eyelids | 15.4 | 2.5 | 4.0 |
| Long and Voluminous Hair | 14.9 | | |
| Tired-looking Eyes | 13.4 | | |
| Gray Hair | 12.3 | | |
| Wrinkles on Cheeks | 11.9 | | |
| Small Eyes | 3.6 | | |
| Thick Eyebrow | 0.0 | | |
| Eyebrow Shape | 2.1 | | |
| Prominent Cheekbones | 0.7 | | |
| Visible Nasolabial Folds | 0.6 | | |
| Prominent Jawline | 0.6 | | |
| Pale Skin | 0.8 | | |
| Serious Expression | 0.1 | | |
| Dark Hair Color | 0.0 | | |
| Soft Lighting | 0.2 | | |
| Tilted Head | 0.3 | | |
| Prominent Ears | 0.1 | | |
| Dark Facial Hair | 2.8 | | |
| Pronounced Makeup | 8.0 | | |
| Thin and Downturned Lips | 5.0 | | |
| Detailed Background | 0.2 | | |
| Low Camera Angle | 0.1 | | |
| *after manual combinations of concepts with $\widehat{\text{PNS}} \geq 5\%$ (Sec. 3.5)* | | | |
| Hairline + Wrinkles on Forehead | 47.8 | 8.2 | 8.2 |
| Glasses + Wrinkles around Eyes | 60.4 | 21.7 | 23.6 |
| Glasses + Hairline | 61.4 | 24.6 | 25.4 |
| Glasses + Hairline + Wrinkles on Forehead | 78.2 | 26.0 | 27.1 |
| Glasses + Wrinkles on Forehead | 59.6 | 24.1 | 27.6 |
| Glasses + Wrinkles on Forehead + Eyes | 65.6 | 26.8 | 28.9 |

This is due to the object-aware generative backbone, which manipulates disentangled representations of objects. For example, cars may appear or disappear (left and middle images) and shift their position, blocking lanes that were previously free. In Stage 2, the VLM produces detailed change captions that describe the point-by-point differences between the original images and their corresponding counterfactual explanations. While most of the caption content is accurate, we observe occasional noise and hallucinations from the VLM, where it describes changes that are not actually present in the images.

**Ablation of Stage 2.** In the baseline, described in Sec. 4.3 ('Ablation of Stage 2'), instead of feeding the LLM with *change captions*, we provide simple captions for all images and counterfactuals. To generate the captions for each image, we use the recent Florence-2 (Xiao et al., 2024), prompted with the 'More detailed caption' task mode. Then, we feed all the captions, along with the classification yielded by the target model $M$, to the LLM, with a prompt shown in Fig. 14. LLM fails to hypothesize that Class '1' is confounded

You are an expert in street/road traffic analysis and driving-scene analysis.
[INST] List how Image-2 differs from Image-1:
- Consider how each element of the image (objects, people, animals, etc.) has changed, considering their sizes, materials, colors, textures, poses, positions, and relative positions
- Consider elements that might have appeared or disappeared in the transition to Image-2 from Image-1
- Consider whether the image or scene as a whole has changed, and if that's the case, explain how
- Omit commonalities; focus only on the differences
- BE RIGOROUS: DO include comparisons that are clearly visible in both images; DO NOT guess or infer details
- Present the changes as a plain-text list, one change per line, in the format <element> → [<specifics> →] → <change> [/INST]
—
Image-1: IMAGE_TOKEN
—
Image-2: IMAGE_TOKEN
—
CHANGES:
road → left lane → was clear, now blocked by a car in the same direction
road → right lane → had an incoming car, now has car in the same direction
road → middle lane → surface more reflective
road → pedestrian crossing → less visible; more faded
traffic lights → leftmost → appeared; green light
middle car → brighter; color more saturated
rightmost streetlight → disappeared
ego-car → dashboard → brighter
—
Image-1: IMAGE_TOKEN
—
Image-2: IMAGE_TOKEN
—
CHANGES:

Figure 12: **Prompts used for stage 2 for BDD.** We prompt the Pixtral VLM for the image change captioning stage. The special tokens 'IMAGE_TOKEN' are replaced by the tokenized images. We use a one-shot in-prompt annotated sample as guidance, where we manually describe the differences between an image (randomly selected) and the obtained counterfactual explanation after Stage 1.

by the presence of vehicles in the left lane. This ablation highlights the importance of fine-grained pairwise comparisons between an image and its counterfactual explanation, as provided by Stage 2, are crucial for the effectiveness of our method.

**Noise from Stage 2.** As we rely on a zero-shot VLM (Pixtral) for change-captioning in Stage 2, we observe substantially higher noise levels compared to CLEVR, where a domain-specific captioning model was available. To quantify this, we ran a small diagnostic study on five randomly selected counterfactual pairs for the biased classifier. Across these samples, we manually counted 14 incorrect change descriptions out of the 35 total described changes (e.g., spurious mentions of changes in streetlight visibility, traffic light color, or added/removed pedestrians), yielding an estimated hallucination rate of approximately 40%.

**Ablation of Stages 1 and 2.** In the baseline, described in Sec. 4.3 ('Ablation of Stages 1 and 2'), we prompt the LLM of Stage 3 to imagine possible biases that the target classifier may suffer. There are thus no counterfactual explanations generation and the LLM does not depend on any information given by the classifier, except for the class meaning (=labels). We show the prompt and output in Fig. 15. As there are

---

You are an expert in image analysis and forensic facial comparison.
[INST] List how Image-2 differs from Image-1:
- Consider how each facial feature has changed, including the eyes, nose, mouth, ears, hair, etc
- Consider whether the facial expression has changed and how this has affected the facial appearance
- Consider whether the pose and camera angle have changed
- Consider whether the lighting, background, and other environmental factors have changed
- Omit commonalities; focus only on the differences
- BE RIGOROUS: DO include comparisons that are clearly visible in both images; DO NOT guess or infer details
- Present the changes as a plain-text list, one change per line, in the format <element> → [<specifics> →] → <change> [/INST]
—
Image-1: IMAGE_TOKEN
—
Image-2: IMAGE_TOKEN
—
CHANGES:
forehead → lines appeared
eyes → darker; less shiny
eyelids → lower → periorbital edema appeared
cheeks → more prominent; lines appeared
nose → slightly wider
nasolabial folds → became apparent; very marked
lips → thinner; darker; less red
—
Image-1: IMAGE_TOKEN
—
Image-2: IMAGE_TOKEN
—
CHANGES:

---

Figure 13: **Prompts used for stage 2 for CelebA.** We prompt the Pixtral VLM for the image change captioning stage. The special tokens 'IMAGE_TOKEN' are replaced by the tokenized images. We use a one-shot in-prompt annotated sample as guidance, where we manually describe the differences between an image (randomly selected) and the obtained counterfactual explanation after Stage 1.

no dependencies anymore on the target classifier, the LLM can only imagine generic biases and fails to find the unexpected left-lane bias. As LLM-generated hypotheses fail to raise the left-lane-vehicle bias, it shows the importance of the counterfactual guidance (Stages 1 and 2) in the GIFT framework

**Stage 3 adaptability to the captions' information.** Our main experiment used a large number of counterfactuals, approximately 150 image pairs, as reported in the paper. This was our initial attempt, without tweaking this number, to ensure fairness. We then conducted an ablation study to determine the minimum number of pairs needed to detect the bias. The results showed that at least 20 pairs are required to reliably identify the bias. The key insights are: using a large number of pairs works well, as the LLM is able to reason effectively and filter out noise. We hypothesize that a smaller LLM might struggle in this scenario. Additionally, unlike CLEVR, where only three pairs were sufficient, the noisier change captions in this case require more pairs to allow the LLM to distinguish the true signal from the noise.

### H.2.1 Stage 4 qualitative samples

Fig. 16 shows image intervention for the 'Dense Traffic in the Left Lane' explanation.

# I  Computational footprint.

In our runtime breakdown, on an A100, for either CelebA or BDD classifier, Stage 1 (OCTET) takes 4.2s/image (batch of 16), Stage 2 (Pixtral) takes 3.5/image and Stage 3 takes 20s (total). Stage 4 takes 1.9s/image (batch of 32).

Each one of the descriptions below (separated by '—') details the content of an image. The descriptions are grouped by the prediction of a binary classifier on their corresponding images: images classified as 0 and images classified as 1. Your task is to consider ALL the descriptions and identify the main factors that could lead the classifier to choose class 0 over class 1, and vice-versa.

The set of description is noisy, and the descriptions may contain irrelevant or contradictory information. Your task is to focus on the most important factors that appear consistently across many instances

-Find testable factors that can be observed and measured in the images (e.g., objects' presence, appearance, arrangement, etc.)

-Present your factors as a bulleted list

Description of images classified as 0:

—

The image is of a highway at night. The sky is dark and the street lights are on. There are two yellow lines painted on the road. On both sides of the road there are rows of parked cars. The cars are all black in color. In the front of the image there is a windshield of a car. The headlights of the cars are on and they are all turned on.

—

Cars are driving on a street at night. The street has white lines on it. There are buildings on both sides of the street. The sky is dark and there are street lights on.

—

. . .

Description of images classified as 1:

—

A car is driving down the street. There are several cars parked on the side of the street next to tall buildings. There is a red car in the middle of the road.

—

This is an image of a highway at night. The sky is completely black. The street lights are on. There are two yellow lines painted on the road. There is a red car in the middle of the street. The headlights of the car are on and it is dark outside. The car is driving on the right side of the road and there are several other cars on the left side.

—

A car is driving down the street at night. There are white lines painted on the street. There is a green light in the distance on the road.

. . .

—

[**Answer**]

- Street Lines and Markings: More detailed and frequent in class 0.
- Parked Cars: More common in class 0.
- Building Appearance: More detailed and diverse in class 1.
- Lighting and Reflections: Brighter and more reflective in class 1.
- Time of Day: More night time in class 0, more varied in class 1.
- Traffic Density: Higher in class 1.
- Perspective: More inside car in class 0, more varied in class 1

Figure 14: **Ablation of Stage 2: Prompt and LLM Response in Stage 3**. The top section shows the prompt used to query the LLM, containing Florence-2-generated captions for each image and its counterfactual, grouped by the target model's predictions. Class '0' denotes *Cannot turn right*, and Class '1' denotes *Can turn right*. The bottom shows the LLM's response, which notably fails to identify that Class '1' is confounded by vehicles in the left lane. This ablation underscores the importance of fine-grained, pairwise comparisons for our method's effectiveness.

---

[**Prompt**] :

You are an expert in analyzing and explaining the behavior of machine learning classifiers. Consider a binary classifier designed to predict whether a vehicle in a driving scene can turn right. This classifier is potentially imperfect. What possible biases or spurious correlations might this classifier have regarding the decision to turn right?

—

[**Answer by Qwen 2.5**] :

- Time of Day: The classifier might learn that certain times of day (e.g., rush hour) are more likely to have restrictions on turning right due to increased traffic.
- Day of the Week: Similar to time of day, the classifier might pick up on patterns where certain days (e.g., weekends vs. weekdays) have different traffic conditions that affect the ability to turn right.
- Urban vs. Rural Areas: The classifier might learn that turning right is more common or less restricted in rural areas compared to urban areas due to differences in traffic density and infrastructure.
- Specific Locations: The classifier might overfit to specific intersections or regions where turning right is either very common or very rare, leading to poor generalization to new locations.
- Weather Patterns: The classifier might learn that certain weather conditions (e.g., rain, snow) make it less likely for vehicles to turn right due to reduced visibility or road conditions.
- Type of Vehicle: The classifier might pick up on correlations between the type of vehicle (e.g., trucks, cars, motorcycles) and the likelihood of turning right. For example, larger vehicles might be less likely to turn right in tight spaces.
- Traffic Light States: The classifier might learn that certain states of traffic lights (e.g., red light) are associated with not being able to turn right.
- Signage: The presence or absence of specific traffic signs (e.g., "No Right Turn" signs) might be overemphasized by the classifier.
- Driver Demographics: If the dataset includes information about the driver, the classifier might pick up on spurious correlations related to the age, gender, or experience level of the driver.
- Driver Intent: The classifier might infer the driver's intent to turn right based on subtle cues that are not directly related to the feasibility of the turn (e.g., the position of the turn signal).
- Dataset Imbalance: If the dataset contains a disproportionate number of examples where turning right is allowed or prohibited, the classifier might be biased towards the more frequent class.
- Sensor Quality: Differences in the quality or type of sensors used to collect data (e.g., cameras, LIDAR) might introduce biases if the classifier learns to rely on artifacts specific to certain sensor types.
- Overfitting to Training Data: The classifier might overfit to the specific characteristics of the training data, leading to poor performance on unseen data. This can manifest as spurious correlations that do not generalize well.
- Pedestrian Presence: The classifier might learn that the presence of pedestrians near the intersection makes it less likely to turn right.
- Bicycle Lanes: The presence of bicycle lanes might be correlated with restrictions on turning right, especially in areas with high bicycle traffic.

Figure 15: **Ablation of stages 1 and 2 on BDD-OIA bias discovery.** In this experiment, we prompt the LLM in Stage 3 to imagine possible biases that the target classifier may have (prompt shown at the top), and display the LLM's response (bottom). Without relying on the target classifier, the LLM only generates generic biases and fails to detect the unexpected left-lane bias. This demonstrates the critical role of counterfactual guidance (stages 1 and 2) in the GIFT framework.

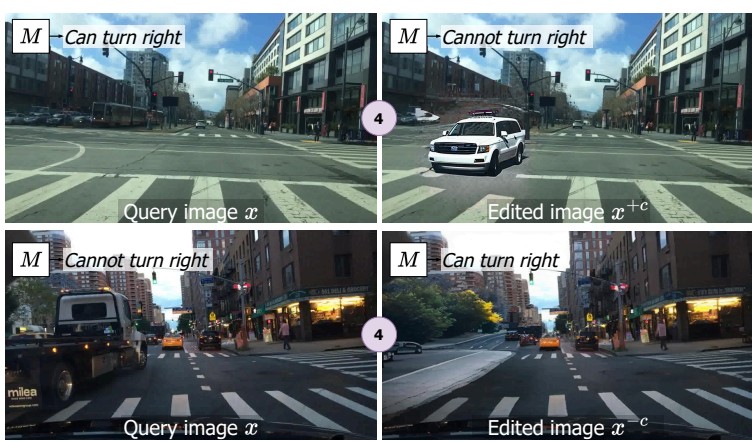

Figure 16: **Image intervention for the 'Dense Traffic in Left Lane' explanation.** The model is biased for vehicles in the left lanes to yield the 'Cannot turn right' output. Left: query images; Right: images $x^{+c}$ and $x^{-c}$ with the added and removed concept.

