# OpenReview forum: "GIFT: A Framework Towards Global Interpretable Faithful Textual Explanations of Vision Classifiers"
_TMLR — Accepted by TMLR_

### Review · Reviewer_kjB2 · 2025-10-30

**Summary Of Contributions:**

This paper introduces GIFT, a novel framework designed to generate textual global explanations by leveraging off-the-shelf components: a counterfactual generator, a large language model (LLM), and an image editing model. The proposed pipeline operates in four key steps: (1) generating a set of counterfactual examples, (2) producing captions for these counterfactuals, (3) deriving candidate global explanations from the captions, and (4) validating the explanations by probing the target model’s behavior under generated interventions. To evaluate the effectiveness of GIFT, the authors conduct comprehensive experiments that showcase that she explanations are able to undercover induced classifications rules.

**Strengths**:
- Novelty and Potential Impact: The paper introduces a unique approach to generating global, textual, and counterfactual-based explanations, which could serve as a foundation for new XAI methods. Beyond interpretability, the flexibility of GIFT opens avenues for applications in reverse engineering, as demonstrated in the experiments.

- Clarity and Rigor: The paper is well-written, with a clear and structured presentation. The related work section is thorough, and the ablation studies address key questions a reader might have. The inclusion of a dedicated limitations section further strengthens the work.

- Explanation Quality Assurance: A major strength is the emphasis on verifying explanation quality, an aspect often neglected in prior work on textual explanations. This rigorous validation enhances the reliability and trustworthiness of the proposed method.

- Scalability and Future-Proofing: GIFT’s instance-based architecture makes it scalable, allowing seamless integration with future advancements in LLMs and image generation models. This adaptability ensures its relevance as underlying technologies evolve.

**Weaknesses**:

- Limited Experimental Scope: The experiments focus exclusively on binary classification, leaving open questions about how GIFT scales to multi-class settings. While computational costs are currently reasonable, the method’s efficiency and effectiveness may degrade as the number of classes increases.

- Overstated Claims: The assertion that GIFT is the "first framework for global, faithful, interpretable explanations for vision classifiers" may be overstated. Prior works [1] have explored similar axioms using discrete representations, and claiming full interpretability and faithfulness is ambitious, given the inherent limitations of counterfactual generation and LLM-based reasoning.

- Faithfulness Concerns: The fourth stage—validating explanations via model probing—does not fully guarantee faithfulness, as it relies on an intervention generation process that may itself be prone to errors or inaccuracies. If the image editing or counterfactual generation fails, the resulting explanations could be misleading. A more nuanced discussion of these dependencies would strengthen the claims.

**Additional Comments:**

References:

[1] Nauta, Meike, et al. "From anecdotal evidence to quantitative evaluation methods: A systematic review on evaluating explainable ai." ACM Computing Surveys 55.13s (2023): 1-42.

[2] Shaham, Tamar Rott, et al. "A multimodal automated interpretability agent." Forty-first International Conference on Machine Learning. 2024.

**Audience:**

Yes

**Audience Explanation:**

The proposed approach introduces a novel class of explanations with, to my knowledge, no direct equivalents in existing literature. The emphasis on hypothesis verification addresses a critical gap in current XAI research, where rigorous validation of explanations remains underexplored. Moreover, the framework’s design opens the door to a wide range of emergent applications, making it a promising contribution to both interpretability and broader model analysis.

**Claims And Evidence:**

Yes

**Claims Explanation:**

While I acknowledge that some claims in the paper—particularly regarding faithfulness and interpretability—may be ambitious, I recognize that these notions are inherently subjective and open to interpretation. If the authors ground their assertions in supporting literature, their perspective can be reasonably justified.
Regarding the experiments, although the relatively low $\hat{PNS}$ scores raise questions about performance, the diversity of tasks and datasets (three distinct datasets were used) provides stronger empirical evidence and helps mitigate these concerns. This breadth enhances the robustness and generalizability of the findings.

**Requested Changes:**

- Potential addition in related works: [2]
- To substantiate the claims, the authors should either:
     -  Support with evidence: Provide studies demonstrating that textual explanations are more interpretable than alternatives listed in Table 1.
Clarify why counterfactual-based methods should be considered fully faithful, referencing existing literature or empirical validation.
     - Mitigate the claims: Revise statements like "We introduce the first framework for global, faithful, interpretable explanations for vision classifiers" to a more precise formulation, such as: "We introduce the first framework for global, textual, and counterfactual explanations for vision classifiers. The faithfulness of the explanations is supported by causality quantification."


- Discuss the potential impact of increasing the number of labels on the method’s computational efficiency, explanation quality, and overall viability. Address whether the current approach can scale effectively or if adaptations would be necessary for multi-class settings.

---

> ### Author Response · Authors · 2025-12-08
> **Answer to Reviewer kjB2**
>
> We thank Reviewer kjB2 for the constructive feedback and positive assessment of GIFT. We are glad for the appreciation of the novelty of the framework, the rigorous verification process, and the clarity of our presentation. Your insights on the method's flexibility and potential for future applications are encouraging. To facilitate tracking changes, updates to the paper have been written in blue.
>
> **1. Limited Experimental Scope (Multi-class settings)**
>
> GIFT is designed to explain specific decision behaviors (e.g., "Why Class A and not Class B?" or "Why Class A and not the rest?"). In a multi-class setting, we apply GIFT in a "one-vs-rest" or "one-vs-one" manner, as noted in Section 3.4. The approach remains viable without modification. We simply target the specific transition we want to understand. The cost scales linearly with the number of *relationships* one wishes to analyze, not necessarily the total number of classes. For example, explaining the top-1 prediction against the runner-up requires only one run of the pipeline. We have added a paragraph detailing this scaling behavior in the Limitations section (Section 5 of the revised paper).
>
> **2. Overstated Claims (Faithfulness and Interpretability)**
>
> We agree that our initial phrasing was too strong, and could be interpreted as claiming full interpretability and faithfulness. We have adopted your suggestions to tone down and mitigate these claims:
> * We changed the title to *“GIFT: A Framework **towards** Global Interpretable Faithful Textual Explanations of Vision Classifiers”*. We also toned down claims throughout the paper (Abstract, Introduction, Conclusion, in particular) accordingly.
> * We argue that natural language is more accessible to humans than saliency maps or feature vectors. We updated Section 3.2 with references, including Nauta et al. [1], to support the claim that textual explanations offer a higher level of user interpretability compared to low-level visual cues.
> * We clarified in Section 3.1 why counterfactual-based methods are generally considered faithful. Unlike surrogate methods, counterfactuals do not approximate the model. They directly probe its decision boundary by identifying minimal input changes that cause a prediction flip. If $M(x') \neq M(x)$, then the difference between $x$ and $x'$ reflects factors the model truly relies on. This makes counterfactual explanations faithful to the model’s behavior by construction, without assuming an auxiliary model or heuristic interpretation layer.
>
> **3. Faithfulness Concerns (Stage 4 dependencies)**
>
> We agree that Stage 4 depends on the quality of the counterfactual interventions, and we expanded this discussion in the revised *Limitations* section (Sec. 5). Importantly, we now provide a more nuanced analysis of how these dependencies affect faithfulness.
>
> First, our causal verification assumes that the editing model modifies only the target concept while leaving other factors unchanged. This assumption is standard in causal-intervention work (e.g., Goyal et al., 2019a). To make this assumption hold as much as possible in practice, we design the editing pipeline to be highly localized: mask-guided text editing restricts modifications to a small spatial region, reducing the risk of unintended global changes.
>
> Second, to assess how well this assumption holds in practice, we manually evaluated intervention quality on CelebA for three fine-grained attributes, both when *adding* and *removing* concepts. We directly inspected edited images and categorized side effects (i.e., editing other concepts) into *no change*, *light change*, and *moderate change*. We also illustrate each category in the new Fig. 9 & 10 (supp. material).
>
> Below is the quantitative summary:
>
> |Concept|Adding: No change| Adding: Light change|Adding: Moderate change|Removing: No change|Removing: Light change|Removing: Moderate change|
> |-|-|-|-|-|-|-|
> |Receding hairline|54%|30%|16%|82%|18%|0%|
> |Neck wrinkles|52%|24%|24%|52%|26%|22%|
> |Wrinkles on forehead |48%|30%|22%|68%|22%|10%|
>
> *Impact on other concept change, after image intervention on CelebA (%). (50 images inspected per setting)*
>
> Across all concepts, most edits fall into the *no change* or *light change* categories (see Fig. 9–10 in the supplement), and crucially, the occasional secondary changes are not systematic: they vary across samples and are not correlated with the manipulated attribute. This greatly limits their ability to produce misleading causal effects in Stage 4.
>
> Overall, while Stage 4 does rely on the quality of counterfactual edits, we now provide a clearer articulation of these dependencies and empirical evidence that interventions are sufficiently precise, and their residual noise sufficiently unsystematic, to support faithful causal validation. We have revised Section G.2 (in the Appendix) to incorporate this discussion.
>
> **4. Related Works**
>
> We thank you for the reference, we have added Shaham et al. [2] (MAIA) to the related work section.

---

### Review · Reviewer_2BHc · 2025-11-10

**Summary Of Contributions:**

This work introduces GIFT, a framework which proposes and verifies
textual explanations for global classification decisions.
The framework makes use of multiple large models, including
(1) a contrastive explanation model to generate local contrastive examples,
(2) a caption-change model (or VLM), which generates textual descriptions of
the local changes introduced by the model in (1),
(3) an LLM, which summarizes all textual descriptions in (3) to obtain global textual descriptions, and
(4) a VQA model to identify examples where the textual description of (3)
applies, and a text-guided image-editing model to add/remove the property
described by the textual description, the two of which are used to verify and
quantify whether the textual descriptions are faithful explanations of the model prediction strategy.

The proposed framework is empirically validated on toy dataset (CLEVR),
as well as two realistic datasets (CelebA and BDD-OIA) of increasing
complexity.


### Strengths

- The work discusses all the limitations inherited by the used models.
- Three datasets of increasing complexity are analyzed.
- Key observations are clearly presented (4.3).
- The verification in Stage 4 is a good approach.
- Figure 1 describes the pipeline very well.

### Weaknesses

- The use of multiple possibly biased and/or error-prone models in the proposed
  framework can lead to error accumulation.
- The required performance for a successful recovery of the pipeline (i.e.,
  failure cases) of the individual stages is not evaluated.
- The computational requirement of the proposed framework seems immense.
- The framework heavily relies on the local contrastive examples generated in
  Stage 1, which are unlikely to capture all data properties, and may require
  an infeasible number of examples.
- ChatGPT 4, a proprietary model, is used in one of the experiments, leading to
  non-reproducible results.

**Additional Comments:**

- I think the comparison with state of the art and ablation studies documented
  in the last paragraphs of 4.3 are quite helpful, and summarize the
  observations well. The caption-change ablation is especially interesting.

**Audience:**

Yes

**Audience Explanation:**

The work combines several large models to test whether our current
state-of-the-art allows a reasonably automated way to provide global
prediction strategies of vision classifiers as textual descriptions.
Thus, it provides an important down-stream task for these models,
which also serves as a benchmark.

**Broader Impact Concerns:**

I do not see any ethical implications of this work that would require a broader
impact statement.

**Claims And Evidence:**

No

**Claims Explanation:**

The work demonstrates the recovery of textual classification rules
in an empirical analysis for each of the three analyzed data settings.
However, each of these setups are only presented with specific models.
The model requirements for this analysis to succeed are not analyzed at all.
For instance, what is the cheapest setup one could use to recover
the classification rules for CLEVR?
Furthermore, it is unclear whether the claim that "no prior knowledge is
required" holds, given that the counterfactual models have been trained for specific semantic concepts, and given that the "true" rule seemingly requires
manual combination of more specific rules.

**Requested Changes:**

- 1. (critical) For Section 4.1, you use ChatGPT 4.
  ChatGPT 4 is proprietary and has undocumented changes all the time.
  Therefore, it is safe to assume that any experiment based on ChatGPT
  cannot be reproduced in the future.
  Even if it may be the most performant model,
  I do not think it is a good idea to use a remote service with a model that
  is prone to change without notice, and may not be available in the near
  future for empirical experiments. Would it be possible to use
  some local LLM? Why have you not used Qwen2.5 72B, like in the CelebA
  setting?

- 2. (critical) How do you make sure the counterfactuals cover all semantic
  concepts? How do you make sure to capture "non-semantic" counterfactuals? It
  could be very likely that some secondary attributes of the image, e.g, focus,
  pixel density, camera model, could be used by the model to classify. Unless
  the counterfactual explanation generator is exhaustive, there is a good
  chance that some unintuitive prediction strategies are missed.
  - Could this be the reason GIFT fails in Table 2 for the red metal object on
    ResNet, while it succeeds for ViT?

- 3. (critical) The last paragraph of 4.3 reads: "GIFT requires no prior
   assumptions about potential biases". I do not think this is entirely true,
   as Stage 1 *does* require prior knowledge (of the counterfactual model)
   to generate counterfactual explanations. If counterfactuals are not
   generated for certain attributes in Stage 1, then GIFT also has no way of
   identifying these.
   - This can simply be addressed by changing the wording to weaken the claim.

- 4. (critical) The number of counterfactual examples required to exhaustively
  discover all relevant rules could be extremely large, depending on the
  complexity of the task. And even if one does generate a lot of examples, the
  contrastive example generator might be biased and miss some important
  changes. As also discussed in 4.3, the Stage 2 might also fail for complex
  data examples, or might also be biased.
  - I think one way to at least partly address this concern would be to
    show how the quality of the identification of rules scales with the number
    of contrastive examples generated.

- 5. (critical) As the quality of the analysis can greatly vary with the models
  used, it would be good to compare different models in the different stages in
  4.1. While we can see that the rules of ResNet seem harder to discover, it
  would also be interesting to see how good the models need to be in order for
  the pipeline to succeed.
  - The conclusion claims that the framework was validated "with diverse models
    at each stage", but there is no direct comparison of models. This would be
    addressed by the suggestion above.

- 6. (critical) The combination of the different attributes makes a lot of sense.
  Is this done only manually, or automatically? If not automatically, do you
  think there is a way to do so?
  - Is the counterfactual generator able to create "combined" attributes?

- 7. (stronger) Related work: GIFT is not compared to counterfactual explanations
  or feature attributions. A simple comment on use of counterfactual
  explanations, and differences to feature attributions regarding GIFT would
  suffices.

- 8. (stronger) Stage 4: The work explains that the most promising descriptions
  are kept based on the PNS. It would be helpful to explain how a good
  threshold is chosen, as you also mention in 4.1 that "appropriate
  thresholding" can help. Additionally, the PNS seems low in 4.2, but it is not
  clear to me at which point a rule can be interpreted as "true".

- 9. (stronger) I really welcome the limitations in Appendix J. Given that this
  pipeline setup seems fragile and the limitations are extremely important for
  this setup, I think it would be much better to include these in the main
  manuscript, before the conclusion. This would also still be well within the
  12 page limit.

- 10. (stronger) While the model succeeds in identifying known biases,
  it would strengthen the work a lot if it could be shown
  to identify previously unknown biases. However, I also understand
  that this might be quite challenging to pull off.

### Minor

- Please add a reference for model-agnostic methods: "reduced fidelity and
  limit the reliability of explanations for complex, high-dimensional data."

- "acting as mere edge detectors" -> "resembling edge-detectors on CNNs"
    - they are not edge detectors, only look like it
- input attribution -> feature attribution
- model agnostic is too broad, and I would not call surrogates "agnostic"
- counterfactual explanations are not just images

- no recent work cited on feature attribution
- no recent work cited on surrogate model attribution

- Eq. (2) the $\forall$ should be an "if"

- Why is Stage 1 linked in the PDF, but not the other stages?

- Fine filter Pearl (2009); Goyal et al. should be citep, not citet

- (Freisleben, 2020) reference is published:
```
Freiesleben, Timo. 2022. “The Intriguing Relation Between Counterfactual Explanations and Adversarial Examples.” Minds and Machines 32 (1): 77–109. https://doi.org/10.1007/s11023-021-09580-9.
```

- M is not well defined. Does it encompass *any* class change?
  - I am assuming the "M(x)" produces the predicted label.
  - For the image pairs on page 4, the "model prediction" can be interpreted as
    the class label, same for (3).
  - In (2), the effect on a specific class is measured by an explicit
    difference of the labels, which is counter-intuitive. It works better when
    using the same notation as in (3), which explicitly checks the prediction
    for some class y, which is the true class.

- in the paragraph before (1), maybe use `:=` rather than `=` to define the
  edited samples

- the image pairs on page 4 are missing an equation number

- use a wide-hat for PNS? It looks like only the N has a hat.

- maybe box figure 5, as it looks like the text is continuing from the last page

- ChatGPT4: reference should be citep, not citet

- 4.1 Main results: The true visual rule was uncovered [in] 11/12 cases.

- 4.1 second to last paragraph: (such as 'Red ~~metal~~ object instead of 'Red [metal] object'

- Qwen2.5 citation "Team" is odd

- Figure 4 is very far away from the paragraph where it is first mentioned.

- Table 6 is mentioned in the manuscript, but it is not mentioned that this
  table is in the appendix. Same for Figure 14.

- Figure 6 was a fun addition, my first quick guess was correct

---

> ### Author Response · Authors · 2025-12-08
> **Answer to Reviewer 2BHc (1/4)**
>
> We thank the reviewer for the thorough and constructive review. We are glad for the recognition of the value of our verification stage, the clarity of our presentation, the use of three datasets with increasing complexity.  We are pleased that the comparison with sota methods and the ablation studies were found helpful. Updates to the revised paper have been written in blue.
>
> ### Weaknesses
>
> **1. Error accumulation.**
>
> A key contribution of our work is the Stage 4 verification mechanism that directly addresses the challenge of multi-model dependencies. We acknowledge that relying on multiple pretrained models can introduce noise and bias, which is precisely why we designed Stage 4 verification to identify which explanations remain faithful to the classifier (as motivated in Sec 4.4).
>
> **2. Required performance for a successful recovery of the pipeline.**
>
> We agree that identifying which components are most critical to GIFT’s success is important. While a full quantitative characterization is difficult across heterogeneous domains, we provide concrete insights into the performance requirements of each stage.
>
> * *Stage 1 (Counterfactual generation)* is the main bottleneck, as all subsequent stages depend on meaningful counterfactual pairs. Counterfactuals must remain in-distribution and flip the classifier prediction with plausible edits. When this holds, the rest of the pipeline proceeds reliably. The two methods we tested (OCTET and ACE) worked consistently across domains, even for complex driving scenes, despite their inherent difficulty. Our response to comment (#4) also discusses how many counterfactuals are needed in practice.
> * *Stage 2 (Change captioning)* is the noisiest stage, especially in real-world settings. In CLEVR, where we use a domain-specific captioner, noise is very low: in a manual inspection of 20 random samples, 17/20 captions correctly described the counterfactual change. In contrast, CelebA and BDD rely on a zero-shot VLM (Pixtral), which produces substantially noisier outputs. A small diagnostic study shows hallucination rates of ~33% for CelebA and ~40% for BDD, consistent with the clutter and fine-grained variation in these scenes. Despite this, we found that the pipeline remains robust to Stage-2 noise. Stage 3 tends to ignore spurious captions because hallucinations do not form consistent patterns across counterfactual pairs, and Stage 4 eliminates any remaining incorrect hypotheses through causal verification. Thus, while Stage-2 noise is real and acknowledged, its effect on the final explanations is mitigated by the design of GIFT.
> * *Stage 3 (LLM explanation induction)* requires prompting the LLM to generate *many* hypotheses to ensure high recall. This design is important: weak or noisy candidates are eliminated by the correlation filter and then by the causal filter in Stage 4. In the CLEVR experiments, we found that Qwen2.5 (72B) and ChatGPT4 performed similarly well (finding 11/12 rules). We additionally tested a smaller Llama-3-8B, which recovers 8/12 rules; this suggests that smaller models may lack the reasoning ability needed to summarize and disambiguate patterns across multiple counterfactual-caption pairs. In practice, we suspect that more advanced reasoning-based LLMs could further improve this stage as we explain in Future Work.
> * *Stage 4 (Causal verification)* is sensitive to the quality of image editing, as PNS depends on edits faithfully reflecting the hypothesized change. For the concepts explored in our experiments, current editing models are sufficient, but more challenging, fine-grained edits (e.g., requiring precise 3D spatial reasoning) remain an open problem for the community (see Limitations section). In specialized domains (e.g., radiography, satellite imagery), domain-specific editing models would likely be necessary.
>
> See also answer #5 below.
>
> **3. Computational requirement.**
>
> We agree that the computational cost of GIFT is significant, and we report detailed runtimes in App. I as well as acknowledge this in the Limitations section. GIFT produces global and textual explanations that are faithful and testable. Those objectives naturally require more computation than traditional methods (e.g., local saliency-based approaches). We believe the computational cost is justified by the depth of insights GIFT delivers, such as revealing global model biases and providing a nuanced understanding of classifier decision boundaries. For example, while our approach takes approximately ~13 minutes to generate the set of hypotheses and then ~6 min/hypothesis, this is still a favorable trade-off compared to methods like OCTET (Zemni et al. 2023), which rely on trained human evaluators to manually *and less systematically* identify similar biases (with 35% failures).
>
> **4. Assumptions on the Stage 1**: See below (#2.1)
>
> **5. Use of  ChatGPT 4**: See below (#1)
>
> **6. Model requirements**: See below (#5)
>
> **7. "No prior knowledge" claim**: See below (#2.1 and #3)

---

> > ### Author Response · Authors · 2025-12-08
> > **Answer to Reviewer 2BHc (2/4)**
> >
> > ### Response to requested change:
> >
> > **1. Use of ChatGPT4.**
> >
> > We agree with the reviewer that relying on closed-source models raises reproducibility concerns. Our initial CLEVR experiments used ChatGPT-4 for Stage 3, but for all real-world experiments (CelebA and BDD) we deliberately switched to the open-source Qwen2.5 model to ensure transparency and replicability.
> >
> > Following the reviewer’s suggestion, we also re-ran the CLEVR experiments with Qwen2.5. The results remain effectively unchanged: 11 out of 12 ground-truth rules are still correctly recovered. The single failure case differs (ChatGPT-4 fails on *red metal object* for ResNet, whereas Qwen fails on *purple object* for ResNet), but overall performance and conclusions are stable. We have updated the paper and Table 2 to include both sets of results.
> >
> > **2.1 Coverage of counterfactual explanations**
> >
> > We thank the reviewer for this insightful remark. By definition, a counterfactual explanation must produce an input $x'$ close to the original image $x$ that flips the classifier’s decision. The counterfactual explanation methods are not trained for specific semantic concepts. Still, if the classifier relies on an attribute that the counterfactual generator fails to easily steer (e.g., camera model) then no counterfactual will be generated that flips the decision, and this limitation is immediately detectable. In our experiments, we did not encounter such cases. Furthermore, if the resulting global explanations are incomplete or fail to capture causal factors, this is reflected in low PNS scores, providing a quantitative signal that the coverage is partial.
> >
> > Overall, this limitation is described in the paper (Limitations section, first paragraph): GIFT's scope depends on the counterfactual generator used in Stage 1 and attributes that cannot be manipulated cannot be discovered. A promising direction for future work is to combine counterfactuals from multiple methods to cover different parts of the explanation space, which could broaden coverage. Despite this limitation, grounding global explanations in local counterfactuals still uncovers biases and causal factors that would be difficult to identify using other methods.
> >
> > **2.2 "Red metal object (ResNet)" failure.**
> >
> > The observed failure is due to Stage 3, where the LLM generated only partial rules instead of the full explanation. We manually inspected 20 random samples from Stage 2 outputs and found that 17 out of 20 correctly reflected the counterfactual change, confirming that Stages 1 and 2 performed as intended.
> >
> > **3. "No prior assumptions" claim.**
> >
> > We agree with the reviewer that the original claim was too strong. Following your suggestion, we have revised the wording in Section 4.3 to make it more accurate and appropriately nuanced.
> >
> > **4. Number of counterfactual explanations.**
> >
> > We agree with the reviewer. This concern is partly addressed in Sections F.4 and H.2. In our main experiments, we used a large number of counterfactual pairs by default to avoid tuning this parameter. Our ablation studies then show how performance scales with the number of pairs.
> > - For CLEVR, only three counterfactual pairs are often enough for the LLM to recover the correct rule. This is due to the low noise in Stage 2, which uses a model trained specifically for image change captioning, and the simplicity of the underlying rules.
> > - For BDD, captions are noisier and the rules are more complex. We find that at least twenty pairs are needed for reliable bias identification. Using more pairs is effective because the LLM can filter out noise, while fewer pairs make the task harder.
> > These results provide a first quantitative view of how rule identification quality varies with the number of generated counterfactuals.

---

> > > ### Author Response · Authors · 2025-12-08
> > > **Answer to Reviewer 2BHc (3/4)**
> > >
> > > **5. Model requirements.**
> > >
> > > We agree that the performance of individual models influences the quality of explanations. GIFT is designed as a *framework*, and our goal in the experiments is to validate that this framework works across fundamentally different domains (faces, driving scenes, compositional images) and tasks (rule recovery, fine-grained concept identification, bias discovery). Our results show that GIFT is robust to the choice of models at each stage:
> > > * For CLEVR, we use domain-specific models to achieve low-noise captions and controlled rule recovery.
> > > * For CelebA and BDD, we use the same pipeline across two very different domains. The only difference is the counterfactual generator (diffusion-based ACE for CelebA, GAN-based OCTET for BDD), which is a plug-and-play component. Otherwise, the pipeline and other models remain identical.
> > > * Despite these differences, all instantiations succeed, demonstrating that the framework is not brittle and does not rely on a particular architecture.
> > >
> > > These results demonstrate that GIFT is robust to the choice of architecture at each stage. While CLEVR benefits from domain-specialized models, generalist models (used in CelebA and BDD) are sufficient to recover meaningful, fine-grained explanations in complex, real-world scenarios. The framework’s modularity allows domain-specific improvements when available, but success does not rely on any particular model.
> > >
> > > While a systematic ablation across all model combinations would be interesting, it is not central to validating the core contribution: the design of a general, flexible, and faithful framework. Based on our experiments, we suggest a practical guideline: use specialized models when available, otherwise generalist models are sufficient.
> > >
> > > **6.1 Exploration of the explanation space with attribute combinations.**
> > >
> > > We agree with the reviewer. Currently, attribute combinations are created manually, as described in Section 3.5. Automating this process using an iterative LLM feedback loop is a promising direction for future work, as discussed in the Limitations and Conclusion sections.
> > >
> > > **6.2 Is the counterfactual generator able to create "combined" attributes?.**
> > >
> > > By definition, the counterfactual generator produces any combinations of changes, but as few as possible, that are needed to flip the classifier’s decision, which often involves multiple attributes simultaneously. Examples can be seen in Figures 8 and 9, where several changes occur in the generated images.
> > >
> > > **7. Comparison with counterfactual explanations and feature attribution methods.**
> > >
> > > Counterfactual explanations and feature attribution methods are *local*: they describe how the model behaves on a single instance $x$. In contrast, GIFT produces *global* explanations that summarize how the model behaves across the entire data domain. We clarify this distinction in the related work section, and Table 1 highlights the different scopes of these approaches.
> > >
> > > We also offer a direct comparison to counterfactual explanations in Section 4.3. Zemni et al. (2023) conducted a user study showing that 35% of participants failed to identify the injected bias in the BDD driving scene classifier (the one we use in Sec. 4.3) when only inspecting counterfactual examples. This illustrates the limitations of manual inspection and the lack of full interpretability brought by counterfactual explanations. GIFT overcomes this by automatically aggregating counterfactual evidence and converting it into clear textual explanations with causal scores (PNS). This makes the bias explicit and consistently detectable, without requiring humans to inspect dozens of counterfactual–original image pairs.
> > >
> > > **8. Interpretation of PNS**
> > >
> > > Choosing a fixed PNS threshold is difficult. In real-world settings (e.g., CelebA or BDD), image classifiers often rely on several interacting factors, as discussed in Section 3.5. Each explanation usually captures only a *partial* causal effect, which naturally leads to lower PNS values. In this context, GIFT provides a *ranked list* of plausible causal factors, which is far more informative than expecting a single high absolute score.
> > >
> > > The ranking is what matters most. In CelebA, the most meaningful attributes consistently appear among the top-ranked explanations. In CLEVR, where ground-truth rules are known, the correct rule is consistently ranked first by both PNS and CaCE. This demonstrates that relative comparison is more reliable than selecting a hard threshold, and that a Top-K evaluation is the most meaningful way to interpret the scores.
> > >
> > > **9. Limitations section**
> > >
> > > We agree with the suggestion and have moved the Limitations section from Appendix J into the main paper (now Section 5).

---

> > > > ### Author Response · Authors · 2025-12-08
> > > > **Answer to Reviewer 2BHc (4/4)**
> > > >
> > > > **10. Identification of unknown biases**
> > > >
> > > > We agree that detecting previously unknown biases is an important goal. This is exactly what the BDD-OIA experiment (Section 4.3) was designed to test.
> > > > * *The injected bias was unknown in practice.* The classifier was trained with a non-intuitive shortcut: linking *“Cannot turn **right**”* to the presence of cars in the **left** lane. This is not a meaningful driving rule and is not something a human would naturally suspect.
> > > > * *Humans and baselines failed to identify it.* LLM-seeded methods, hypothesis-driven approaches, and manual inspection of counterfactuals all failed to recover this bias (Table 4). This confirms that the bias was effectively “unknown” from a human and methodological standpoint.
> > > > * *GIFT uncovered it without prior assumptions.* By aggregating local causal signals, GIFT surfaced *“dense traffic in the left lane”* as a top global explanation, with strong causal scores (CaCE 45%, PNS 47%). This is exactly the hidden shortcut embedded in the model.
> > > >
> > > > This directly demonstrates GIFT's ability to detect previously unknown biases.
> > > > A similar phenomenon appears in the CelebA experiment: GIFT finds that the ‘Glasses’ concept alone has a 16% probability of being a necessary and sufficient cause of the ‘Old’ label. Since Glasses correlates with Old at over 20% in the CelebA training data, this strongly suggests that GIFT is also uncovering real, previously unreported dataset-driven shortcuts in more natural settings.
> > > >
> > > >
> > > > ### Minor
> > > > We thank the reviewer for the precise and constructive remarks. All minor comments and suggested improvements have been addressed in the revised version.
> > > >
> > > > #### CaCE definition as a difference of classification labels:
> > > > We agree that our original equations (2) and (3) were misleading.   In the paper, we changed equation (2) to the theoretical definition of CaCE from the original paper (Goyal et al 2019a). In this equation, a major difference between CaCE and PNS appears: CaCE is not dependent on any given class $y$, contrary to PNS. It measures in a more general sense whether a concept changes the classification outcome. We mention this specificity of CaCE in the paper, below equation (2): "$\text{CaCE}_{c_e}$ measures how the addition or removal of ${c_e}$ induces any class change." In the binary case for instance, CaCE reveals whether $c_e$ induces class=1 but also whether it induces class=0 depending on the sign of CaCE as detailed right below equation (3).
> > > >
> > > > To avoid any ambiguity, we also updated in the paper our equation (3): we now only give the empirical version of CaCE in the binary case, with a well defined binary classifier M, with classes 0 and 1.
> > > >
> > > > We explore further the links and differences between CaCE and PNS (due to this independance on y for CaCE) in Appendix C.3.

---

### Review · Reviewer_ZM9U · 2025-11-26

**Summary Of Contributions:**

GIFT proposes a 4-stage framework for explaining vision classifiers that combines: (1) local counterfactual generation, (2) vision-language model translation to text, (3) LLM aggregation into global hypotheses, and (4) causal verification through image interventions. The framework aims to produce explanations that are simultaneously global, interpretable, faithful, and textual—addressing limitations of existing methods. Evaluation on CLEVR, CelebA, and BDD-OIA demonstrates the framework's ability to uncover classification rules and biases.

**Audience:**

Yes

**Audience Explanation:**

The authors address an important application

**Claims And Evidence:**

No

**Claims Explanation:**

Mainly yes.
The claim of "Causal" verification rests on the assumption that the image editing model (Stage 4) changes only the target concept and nothing else. Can you verify this is actually the case, eg by checking intervention quality with a VQA model? Also, do you make sure edited images remain in-distribution. This seems an implicit assumption.

**Requested Changes:**

Experiments have quite a limited scope. Can you add a data set with multi-class classification and a more complex real world task, e.g. the knee radiography daatset used in [1-3]. I would also be interested in how [3] performs as a related baseline from the shortcut detection literature (which should be discussed as related work).

You mention "occasional noise and hallucinations from the VLM, where it describes changes that are not actually present in the images" - can you quantify how often this occurs and the effect of performance?

Can you provide quantitative metrics for explanation completeness and accuracy?

[1]  DeGrave, Alex et al.. AI for radiographic COVID-19 detection selects shortcuts over signal. Nature Machine Intelligence, 3(7):610–619, 2021.

[2] Adebayo, Julius et al. Post hoc explanations may be ineffective for detecting unknown spurious correlation. In International Conference on Learning Representations, 2022

[3] Kuhn, Lukas, et al. "Efficient unsupervised shortcut learning detection and mitigation in transformers." Proceedings of the IEEE/CVF International Conference on Computer Vision, 2025.

---

> ### Author Response · Authors · 2025-12-08
> **Answer to Reviewer ZM9U (1/2)**
>
> We thank Reviewer ZM9U for the constructive feedback. We are pleased that the reviewer finds the application important and that the paper is mainly supported by convincing and clear evidence. Updates to the revised paper have been written in blue.
>
> **1. Image editing model assumptions**
>
> We agree that our causal verification relies on the assumption that editing introduces or removes only the target concept while leaving other factors untouched. This assumption is standard in prior causal-intervention work (e.g., Goyal et al., 2019a), and stated explicitly in Sec. 4.3.
> To make this assumption as realistic as possible in practice, we designed the editing procedure to minimize unintended changes: using *local masks* and *mask-guided* text-based editing tightly constrains where and how the image can be modified. This significantly reduces the chance of altering unrelated attributes and helps ensure that the resulting counterfactuals remain valid interventions for causal testing.
>
> To assess how well this assumption holds in practice, we manually evaluated intervention quality on CelebA for three fine-grained attributes, both when *adding* and *removing* concepts. Instead of relying on a VQA model which may introduce its own biases, we directly inspected edited images and categorized side effects (i.e., editing other concepts) into *no change*, *light change*, and *moderate change*. We also illustrate each category in the new Fig. 9 & 10 (Appendix). Below is the quantitative summary:
>
> |Concept| Adding: No change | Adding: Light change | Adding: Moderate change | Removing: No change | Removing: Light change | Removing: Moderate change |
> |--|-| ---| -| ---| ---| ---|
> |Receding hairline|54% |30% |   16% |82% |18% |0% |
> |Neck wrinkles|52% |24% |24% |52% |26% |22% |
> |Wrinkles on forehead|48% |30% |22% |68% |22% |10% |
>
> *Impact on other concept change, after image intervention on CelebA (%). (50 images inspected per setting)*
>
> Overall, the edits remain tightly focused on the target concept: 'no change' or only 'light' changes account for the clear majority of cases, especially when removing concepts. Importantly, when secondary changes occur, we observe that they are inconsistent across samples (e.g., concept A may occasionally co-appear with C, D, or E), meaning these side effects are not systematically correlated with the manipulated attribute and therefore do not bias causal conclusions.
>
> Finally, as suggested, we revisited our assumption that edited images remain in-distribution. We now make this assumption explicit in Sec. 4.3 and highlight qualitative evidence across domains: CLEVR edits stay perfectly in-distribution (Fig. 2), CelebA edits remain natural (Fig. 3), and BDD edits, although naturally more challenging, are still visually coherent (Fig. 14 in the Appendix). The section G.2 in the Appendix has been revised to incorporate this new analysis.
>
> **2. Experimental scope and request for additional dataset.**
>
> We did our best to cover diverse domains and use cases across three datasets, each representing a different level of visual and semantic complexity:
> * CLEVR: a controlled, compositional environment with full ground truth. We evaluate *12 different classifiers* (ConvNets and ViTs), enabling systematic rule-recovery experiments and quantifying explanation accuracy.
> * CelebA: real-world images with *fine-grained visual attributes*. This setting stresses GIFT’s ability to extract subtle, interpretable rules from natural images.
> * BDD: real-world, *multi-object*, spatially complex driving scenes. This is arguably the closest analogue to the “complex real-world task” requested by the reviewer. Importantly, on BDD, the "right-turn" classifier is trained with a shortcut (left-lane vehicle bias), which prior methods failed to systematically uncover. GIFT identifies this shortcut robustly, demonstrating both practical utility and scalability.
>
> Together, these datasets cover compositional reasoning, fine-grained attribute analysis, and complex scene understanding, and we use them to illustrate different capabilities of the GIFT framework (recovering missing class labels, identifying fine-grained attributes, identifying spurious shortcuts).
>
> GIFT is designed to explain specific decision behaviors (e.g., "Why Class A and not Class B?" or "Why Class A and not the rest?"). In a multi-class setting, we apply GIFT in a "one-vs-rest" or "one-vs-one" manner, as noted in Section 3.4. The approach remains viable without modification. We agree that extending GIFT to medical imaging (e.g., knee radiography) would be valuable future work. Yet, we chose not to include such a dataset in this version due to time and resource constraints. Overall, the diversity of our experiments, the variety of architectures tested, and the range of downstream tasks (classification recovery, concept identification, bias discovery) already provide strong evidence that the GIFT framework generalizes beyond any single model family or dataset.

---

> > ### Author Response · Authors · 2025-12-08
> > **Answer to Reviewer ZM9U (2/2)**
> >
> > **3. Related work [3] (Kuhn et al. 2025).**
> >
> > Thank you for pointing to this related work. We added a discussion of [3] in the related work section. We do not include it as a baseline because it addresses a different problem and relies on assumptions that do not apply to our setting.
> > Method [3] detects shortcut features by analyzing internal model activations. It clusters patch-level representations, extracts prototypes, and then uses a VLM to interpret these prototypes. This requires full access to the model’s hidden features and is limited to the shortcuts that appear in these activations. In contrast, our method does not access model internals and instead relies only on counterfactual evidence at the input-output level. We also aim to explain both "correct" and "incorrect" predictions, not only detect shortcut failures. Finally, our contribution includes causal verification using CaCE and PNS through controlled image edits, which allows us to quantify completeness and coverage of concepts.
> >
> > **4. VLM noise and hallucinations.**
> >
> > In the case of CLEVR, we manually inspected 20 random Stage-2 outputs and found that 17 out of 20 correctly described the counterfactual change. This confirms that, when using a domain-specific change-captioning model, Stages 1 and 2 operate with very low noise.
> >
> > For CelebA and BDD, where we rely on a zero-shot VLM (Pixtral), we observe significantly higher noise levels, as visible in Figure 8 and 9 in the Appendix. To quantify this, we ran a small diagnostic study:
> > - CelebA (Young/Old classifier): across 5 random samples, we manually counted 18 wrong change descriptions out of 55 total described changes (e.g., wrong descriptions of changes in pose, background, hair volume, ...), yielding a total of ~33% hallucination rate.
> > - BDD (turn-right classifier with left-lane bias): across 5 random samples, we manually counted 14 wrong change descriptions out of the total 35 captions (e.g., streetlight visibility, traffic light color, added/removed pedestrians), yielding a total of ~40% hallucination rate.
> >
> > This confirms that zero-shot change-captioning is noisy, particularly in the more complex driving scenes. We revised the paper appendix accordingly (Sections G.2, H.2).
> >
> > Despite this, the overall pipeline remains robust to imprecisions stemming from Stage 2:
> > * *Stage 3 (LLM rule induction)* tends to ignore spurious captions because they do not form recurring patterns across multiple counterfactual pairs.
> > * *Stage 4 (causal verification)* provides a strong safety net: any hypothesis derived from a hallucinated caption fails to produce consistent causal effects and is therefore filtered out.
> > Thus, while VLM noise is real, especially in complex scenes, its impact on the final explanations is mitigated by the multi-stage design of our framework.
> >
> > **5. Quantitative metrics for explanation completeness and accuracy**
> >
> > We quantify both completeness and accuracy/faithfulness using two causal metrics introduced in Sec. 3.4: *Probability of Necessary and Sufficient Cause (PNS)* and *Causal Concept Effect (CaCE)*.
> >
> > PNS evaluates whether a hypothesized concept is *both necessary and sufficient* to induce the classifier’s decision under controlled edits. High PNS indicates that removing the concept reliably flips the prediction (necessity) and adding it restores the prediction (sufficiency). This jointly captures the *accuracy* of the explanation (the concept truly drives the classifier’s behavior) and its *completeness* (the explanation accounts for the relevant causal factor rather than a partial or spurious correlate).
> >
> > In the controlled CLEVR setting, where a single ground-truth rule is known, we provide a direct quantitative accuracy evaluation: GIFT correctly recovers the true rule in 11 out of 12 cases (Table 2). This validates that high PNS/CaCE values correspond to correct, complete explanations in a setting where ground truth is available.

---

> > ### Comment · Reviewer_ZM9U · 2025-12-27
> >
> > Thank you for your response. If  GIFT cannot be readily applied to a real-world datasets and applications, I believe it is important to tone done the contributions and explicitly acknowledge and discuss this limitation. It should be made clear to the reader that the approach is designed for binary classification of natural images such as persons and city scenes only.

---

> > > ### Comment · Reviewer_ZM9U · 2026-01-16
> > >
> > > Dear authors,
> > >
> > > it would be great if you could respond to this, so that I can make my recommendation.
> > >
> > > Thanks.

---

> > > > ### Author Response · Authors · 2026-01-16
> > > > **Answer to Reviewer ZM9U #2**
> > > >
> > > > We thank the reviewer for their comments and engagement with the paper.
> > > >
> > > > We have explicitly revised the manuscript to reflect the boundaries you highlighted. Specifically, we have updated the Abstract, Introduction, Section 4 (Experiments), Limitations and Conclusion to state that our empirical validation focuses on binary classification tasks within natural image domains (compositional synthetitic images, face images, and complex driving scenes). We have removed any claims that could suggest immediate "out-of-the-box" applicability to new specialized domains (like medical imaging) where standard generative models may currently struggle.
> > > >
> > > > To explicitly capture the nuance between current validation and architectural design, we have also added the following clarifications:
> > > > - Binary vs. Multi-class: We note that while our experiments focus on binary decisions, the framework logic supports multi-class problems via standard "One-vs-Rest" decompositions (p6 and Limitations).
> > > > - Domain Dependencies: We clarify that the current restriction to natural images stems from the availability of off-the-shelf generative models rather than the framework's logic itself (Limitations).
> > > >
> > > > We believe these changes accurately reflect the strengths of the paper while being fully transparent about the conditions required to port the method to new domains. We hope this clarification and the corresponding revisions address the reviewer's concerns.

---

### Decision · Action_Editor_b6Qr · 2026-02-02

**Recommendation:** Accept as is

**Audience:**

Yes

**Audience Explanation:**

While the work will be of particular interest to the TMLR audience in interpretable and explainable AI, what distinguishes it is the hypothesis verification step, which identifies explanations that remain faithful to the classifier. This aspect remains underexplored in the literature and may stimulate further research in this direction.

**Claims And Evidence:**

Yes

**Claims Explanation:**

After revision, there is consensus among the expert reviewers that the paper meets TMLR standards. Notable improvements over the initial submission include enhanced reproducibility through the use of a local model instead of ChatGPT-4, improved quantitative evaluation of explanation completeness and accuracy via PNS quantifiers, clearer delineation of scope, and explicit acknowledgment of current limitations.